# Pluvial and potential compound flooding in a coupled coastal modeling framework: New York City during post-tropical cyclone Ida (2021)

Shima Kasaei[1], Philip M. Orton[1], David K. Ralston[2], John C. Warner[3]

[1]Stevens Institute of Technology, Hoboken, NJ, 07030, USA

[2] Woods Hole Oceanographic Institution, Woods Hole, MA, 02543, USA

[3] U.S. Geological Survey, Woods Hole Coastal & Marine Science Center, Woods Hole, MA, 02543, USA

*Correspondence to*: Shima Kasaei (Skasaei@stevens.edu)

**Abstract**

Many coastal-urban areas are prone to extreme pluvial flooding due to limitations in stormwater system capacity, with the additional potential for flooding to be compounded by storm surge, tides, and waves. Understanding and simulating these processes can improve prediction and flood risk management. Here, we adapt the Coupled Ocean-
Atmosphere-Wave-Sediment Transport modeling framework (COAWST) to simulate pluvial flooding from post-tropical cyclone Ida (2021) in the Jamaica Bay watershed of New York City (NYC). We modify the model to capture the volumetric effects of rainfall and parameterize soil infiltration and a stormwater conveyance system as a drainage rate. We generate a spatially continuous flood map of Ida with Root Mean Square Error (RMSE) of 20 cm when compared to high water marks, useful for understanding Ida's impacts and subsequent mitigation planning. Results
show that over 23 km$^2$ and 4621 buildings were flooded deeper than 0.3 m during Ida. Sensitivity analyses are used to study the broader risk from events like Ida (pluvial flooding) as well as potential compound (pluvial-coastal) flooding. Spatial shifting of the storm track within a typical 12-hour forecast uncertainty reveals a worst case scenario that increases this flooded area to 62 km$^2$ (5907 buildings). Shifting Ida's rainfall to coincide with high tide increases this flooded area by 1 km$^2$, a relatively small change due to the lack of significant storm surge. The application of
COAWST to this storm event addresses a broader goal of developing the capability to model compound pluvial-coastal flooding by simultaneously representing coastal storm processes such as rain, tide, waves, erosion, and atmosphere-wave-ocean interactions. The sensitivity analysis results underscore the need for detailed flood risk assessments, showing that Ida, already NYC's worst rain event, could have been even more devastating with slight shifts in storm track.


**Key words:** Modeling, Pluvial flood, Compound flood, Ida, Jamaica Bay, New York City

## 1. Introduction

Coastal regions offer numerous socio-economic and ecological advantages to humans yet face an increasing susceptibility to the detrimental impacts of coastal storms and rising sea levels. Such disturbances precipitate a cascade
of geomorphological and hydrodynamic changes along shorelines, defined by intense wave action, coastal inundation,

erosion, and strong currents, that pose severe threats to human life. An increase of flooding is anticipated due to global warming influences that raise sea levels and augment the atmospheric capacity for moisture retention, thereby increasing the frequency of intense rainfall events (Slater and Villarini, 2016; Trenberth, 2011; Zhu, 2013). In the United States, coastal counties, which house nearly 40% of the population, face substantial risks of flooding due to their low-lying, densely populated, and often extensively developed nature (National Oceanic and Atmospheric Administration, 2023). As such, communities and authorities in flood-prone areas are increasingly confronted with the need to prepare for or respond to these escalating risks (Zinda et al., 2021).

Understanding the underlying mechanisms of these coastal processes and improving predictive models is imperative for informed coastal management and storm preparedness. These improvements can upgrade emergency management by capturing more aspects of coastal storm hazards in forecasts. They can also enable planners and coastal managers to increase awareness, minimize loss of life and property, and support sustainable development by better managing coastal resources.

Coastal flooding can arise from tides, storm surges, waves and intense precipitation, with the latter influencing direct runoff (pluvial) or increased river discharge (fluvial). The concurrence of these flood drivers, termed compound flooding, amplifies the potential for inundation of low-lying coastal areas, surpassing the risk associated with each mechanism in isolation (Wahl et al., 2015). Recent studies of compound pluvial-coastal flooding have primarily been statistical, bivariate copula, assessing the joint probability of water level variability due to the tides or storm surge and rainfall (Zellou and Rahali, 2019; Jane et al., 2022; Kim et al., 2022). A bivariate copula is a statistical tool used in compound flood research to capture and analyze the joint dependance of variables such as river discharge and coastal water levels (Genest and Favre, 2007). An investigation of historical data showed a higher possibility of co-occurrence of storm surge and heavy rainfall for the Atlantic and Gulf coasts in comparison with the Pacific coast (Wahl et al., 2015). The same study found an increase in such compound events in some coastal cities including New York City (NYC) over the past century due to a shift in weather patterns.

Modeling tools have been a limiting factor in research on compound flooding, as in the field of flood modeling, development of coupled hydrologic, hydraulic, and coastal models is rare (Santiago-Collazo et al., 2024). While coastal total water level modeling, forecasting and hazard assessments are common, Hydrologic and Hydraulic models (H&H models) are rarely applied to these problems and when they are, it is typically in a one-way coupled approach (Santiago-Collazo et al., 2019). A key challenge for development of these models is the difficulty of obtaining the comprehensive data required for stormwater modeling, especially for older neighborhoods and private properties. A review of flood modeling methods underscored a concerning neglect of the pluvial flood driver in compound flood risk assessment, showing the essential need for more comprehensive models (Bulti and Abebe, 2020). This is particularly concerning for highly urbanized coastal areas where dense populations, infrastructure, and distinct hydrology exacerbate flood risks.

A recent review of coastal compound flooding recommended development of modelling frameworks that can comprehensively represent the dynamic Earth systems driving compound flooding (Green et al., 2024). The Coupled Ocean-Atmosphere-Wave-Sediment Transport (COAWST) coastal system model couples the Regional Ocean Modeling System (ROMS) with atmospheric, wave, and sediment transport models, enabling the simulation of

interactions and feedbacks between these systems (Bao et al., 2022). However, the COAWST model has heretofore lacked the capability to simulate pluvial flood processes. While models such as the Hydrologic Engineering Center's Hydrologic Modeling System (HEC-HMS) (Peters, 1998), Interconnected Channel and Pond Routing (ICPR), which is now called StormWise (Schroeder et al., 2022), and Storm Water Management Model (SWMM) (Rossman and Huber, 2016) focus on inland hydrological and stormwater system processes, COAWST also incorporates three-dimensional hydrodynamics which is important for accurately predicting baroclinic and stratification effects on storm tides in coastal and estuarine areas (Orton et al., 2012; Ye et al., 2020).

This study improves COAWST to enable pluvial and compound (pluvial-coastal) flood simulations. We simulate Ida's flooding in the Jamaica Bay watershed of New York City in 2021, including exploring different counterfactual scenarios of a shifted storm track causing more intense rainfall and a shifted storm timing causing the rain to peak at high tide to cause compound flooding. Section 2 outlines the methodology behind the coupled modeling framework, which simulates pluvial flooding by accounting for the volumetric effects of rainfall and a simplified representation of urban stormwater drainage. In Section 3, the model is calibrated and validated using flood depth observations and a sensitivity analysis of flooding in three counterfactual scenarios is presented. Section 4 has a discussion of the study's findings and their broader implications, as well as the model's predictive strengths and areas of potential improvement. Lastly, Section 5 provides a summary of the key conclusions.

## 2. Methods

### 2.1. Storm event and study site

Tropical storm Ida formed in the western Caribbean Sea southwest of Jamaica near 23 Aug 2021, 12:00 UTC. It first hit western Cuba as a Category 1 hurricane and later transformed into an extratropical low. The tropical cyclone travelled northwest and strengthened as it entered the Gulf of Mexico. Sea surface temperatures near 30ºC led to continued intensification and the storm became a Category 4 hurricane before making landfall at the Louisiana coast near Port Fourchon on 29 Aug 2021, 16:55 UTC. After landfall, the storm continued to travel across the United States and resulted in severe rainfall and deadly flooding in Pennsylvania, New Jersey, New York, Connecticut and Maryland (Beven et al., 2022). Ida caused the 5[th] wettest day in NYC history with over 17.8 cm (7 inches) rain in total (at Central Park) and set the single-hour rain record at 8 cm (3.15 inches) (Government, 2023; Fema, 2023; Laboratory, 2021). Smith et al. (2023) documented the extreme short duration rainfall and the flooding caused by Ida in eastern Pennsylvania and New Jersey, emphasizing the role of supercell thunderstorms. At NYC, the storm's sustained rainfall overwhelmed stormwater conveyance systems (sewers) and turned streets into rivers in many places, including severe flooding in all five boroughs (Finkelstein et al., 2023).

The coastal embayment and surrounding watershed of Jamaica Bay is a part of New York City (NYC) that experienced extensive pluvial flooding during Ida (Fig. 1). Over 2.8 million people live in Jamaica Bay watershed (NYC-DEP, 2018), and many are situated within range of a realistic 5-meter coastal flood (Orton et al., 2015). Jamaica Bay has an area of 72 $km^2$, encompassing over 15 $km^2$ of marshes and 4.6 $km^2$ of intertidal unvegetated areas, hosting a wide range of habitats and wildlife and offering a variety of recreational opportunities (Orton et al., 2020a; Orton et al., 2020b; Swanson et al., 2016). Ida's rain averaged across the Jamaica Bay watershed was 71 mm (2.8 inches) over 3 hours, which corresponds to a 10-year return period (between 5 and 25 years) based on NOAA Precipitation

 Frequency Data Server (Station ID 30-5803). However, the maximum intensity (Fig. 3) reached 70 mm/hour, or a one-hour rain event with a 50-year return period (2.76 inch/hour) based on the same NOAA data (National Weather Service). In contrast, the prior major disaster in the region of Hurricane Sandy (2012) was predominantly a coastal flooding event that severely impacted neighborhoods surrounding Jamaica Bay and spurred major efforts focused on strengthening coastal defenses (USACE, 2022)).

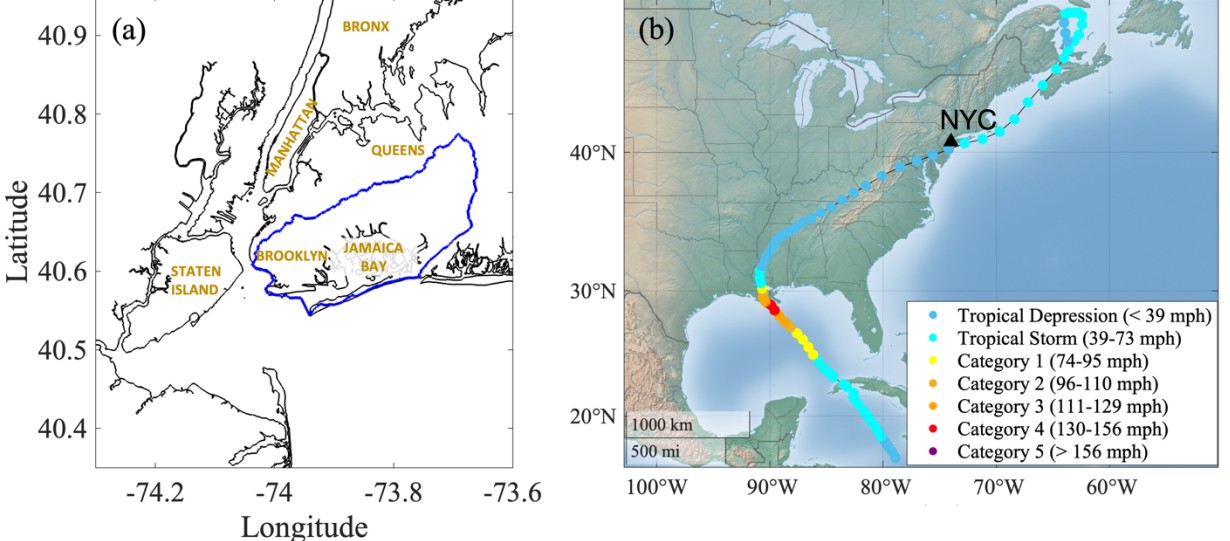

**Figure 1.a) Jamaica Bay location in New York City and its watershed area (shown with blue line), b) Ida storm track (the "colorterrain" base map from MATLAB is applied, hosted by Esri®)**

### 2.2. Modeling

The COAWST modeling system integrates multiple components to comprehensively simulate coastal system interactions (Warner et al., 2010). These components include models for the ocean, atmosphere, surface waves, sediment transport, a coupler to exchange data fields, and a re-gridding method. This gives COAWST capabilities that are not typically available with urban hydraulic-hydrologic models (see Sect. 4 for examples), which are used to design improvements to water infrastructural systems and for predicting urban water cycle processes. COAWST is well suited to represent interactions between the ocean and atmosphere, such as sea-surface temperature, or wave dynamics such as wave generation and propagation.

#### 2.2.1. Hydrodynamic Model

The ocean model in COAWST is the Regional Ocean Modeling System (ROMS), which is a three-dimensional, free-surface, terrain-following hydrodynamic model (Shchepetkin and Mcwilliams, 2005; Haidvogel et al., 2008). It employs finite-difference approximations of the Reynolds-Averaged Navier-Stokes (RANS) equations under the hydrostatic and Boussinesq assumptions (Haidvogel et al., 2000; Chassignet et al., 2000). In ROMS, the

hydrostatic primitive equations are approximated through the utilization of boundary-fitted, orthogonal curvilinear coordinates on a staggered Arakawa C-grid. The vertical dimension uses stretched terrain-following coordinates.

We adapt the ROMS component for this study to account for rain as a volumetric addition to the grid cell, enabling rain-on-grid over water or dry land (COAWSTv3.8). Rainfall is directly integrated into the governing equations in the model. The vertical momentum equation is modified to include the vertical displacement of the water level due to the volume of rain, affecting the free surface and water column dynamics. Additionally, the continuity equation includes a source term to account for the rainfall's volumetric contribution. This approach aligns with other studies (Dresback et al., 2023; Santiago-Collazo et al., 2024), where rainfall is incorporated by modifying the governing equations. The rain rate is included as a spatially and temporally varying meteorological forcing variable. The model can also implicitly account for spatially varying floodwater infiltration and flow into stormwater sewers with drain rates that are subtracted from the rain rate. The drain rate is always negative (a volume sink representing stormwater system and infiltration) while the rain rate is always positive (a volume source), and the net rate of volume change (precipitation-drain rate) can be negative when it is locally greater than the rain rate. For the Jamaica Bay watershed simulations, we assume a uniform, constant drain rate for the land area and use it as a calibration parameter, as described in Sect. 2.4. Volume of the rain that is removed from the domain with this drain term could be routed in the ocean with additional modifications to model, but this is left for future work as it is not an essential component for this study.

### 2.2.2. Model domains, nesting and setup

A nested modeling application is used for Ida, with an existing larger scale coastal and estuarine domain providing boundary conditions for a higher-resolution Jamaica Bay domain. The model grid of the Jamaica Bay watershed has 818 x 734 cells, averaging 46 x 51 m in size with slight variation across the domain, and 8 uniformly spaced vertical sigma layers. The bare-Earth Digital Elevation Model (DEM) for model bathymetry is created by merging three datasets (with descending order of preference): National Park Service data (Flood, 2011), NOAA-NCEI one-ninth arc-second resolution DEM (CIRES, 2014b) and one-third arc-second resolution bathymetric data (CIRES, 2014a). Fig. 2b). We use a spatially varied bottom roughness, in which the quadratic bottom roughness coefficient was calculated based on the land cover (Fig. 2d). The Coastal Change Analysis Program (C-CAP) is a nationally standardized effort by the NOAA Office for Coastal Management that provides raster-based inventories of land cover for U.S. coastal regions, derived from the analysis of remotely sensed imagery to ensure consistency over time and geography (National Oceanic and Atmospheric Administration, 2016). We use the corresponding Manning number for each value of C-CAP (ranges from 0.02 to 0.13) (Mattocks and Forbes, 2008); further we calculate the Zo (bottom roughness) and the quadratic bottom roughness coefficient (which is being used in the model) consecutively. The internal (baroclinic) time step for the simulation is 2.5 seconds, each with 20 external (barotropic) time steps. For wetting and drying, the minimum depth to allow flow out of the cells, Dcrit, is set to 5 cm (Warner et al., 2013). The Ida simulation commences from a state of rest and temperature and salinity are initialized as spatially constant values, although the full 3-D salt and temperature fields could be utilized to capture baroclinicity and stratification effects on storm tides. The open boundaries of the nested model (are shown in Fig. 2a with black rectangle) are set using Chapman conditions for the free surface, Flather conditions for 2D momentum, and radiation conditions for 3D

momentum, and the gradient condition is applied for salinity and temperature, effectively holding them constant within the domain (Marchesiello et al., 2001). Since the spin up is only for velocity and water level, it only requires hours to stabilize.

The regional parent simulation uses a larger scale grid that includes the Hudson River estuary and surrounding coastal region (Figure 2a), as described by Ralston (2022). The open boundaries of the regional parent model are forced with tidal water levels and currents were extracted from the ADCIRC database (Mukai et al., 2002). Additionally, subtidal water levels calculated from observations at the NOAA tide gauges at Sandy Hook (NJ; NOAA station 8531680) and Kings Point (NY; NOAA station 8516945) (Figure 2c) were added to the boundaries in New York Bight and western Long Island Sound, respectively. In addition, we simulate the influences of local wind stress and barometric pressure changes on the regional storm. This combination of boundary conditions and in-domain forcing allows for a more accurate representation of both local and regional storm surge effects. Atmospheric forcing is from the North American Mesoscale Forecast System (NAM) 12-km analysis product. Simulations in the regional model are run for the period 10 Aug 2021 to 10 Sep 2021 to allow for model spin-up prior to Ida hitting NYC. Evaluation of this larger-scale model against previous observations of water level, currents, and salinity are reported with skill metrics in Ralston (2022).

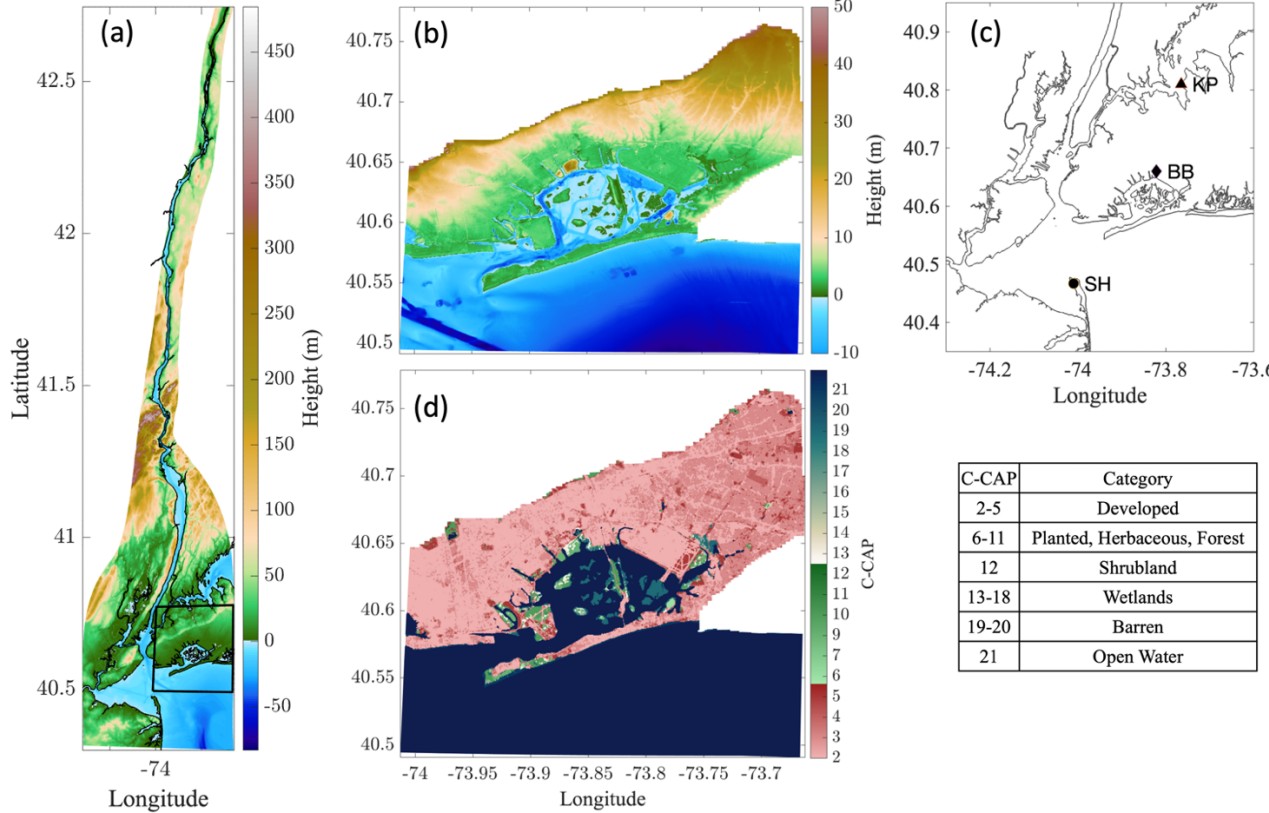

**Figure 2. a) Regional model DEM (m) and the Jamaica Bay model location (black rectangle), b) Jamaica Bay model DEM (m) covering the Jamaica Bay watershed (nested model), c) Tide gauges at Kings Point (KP), Bergen Basin (BB) and Sandy Hook (SH), d) Coastal Change Analysis Program (C-CAP) values and the land-cover categories.**

## 2.3. Application to post-tropical cyclone Ida

The model is applied to investigate Ida impacts for the Jamaica Bay watershed. All simulations for the nested model are initiated with a model time of 29 Aug 2021, 00:00 UTC, when the storm just passed Cuba, and simulations last for five days to 03 Sep 2021, 00:00 UTC. Ida generated only pluvial flooding, since the hour of the most extreme rainfall occurred between low and high tide and the non-tidal anomaly peaked at 0.6 m. At this instance, the observed water level at the Stevens Institute's tide gauge at Bergen Basin in Jamaica Bay (Fig. 2) was 0.21 m NAVD88, which is not enough to cause coastal flooding. This water level is far below the street level, which is 3.17 m NAVD88, and therefore should not block outflow through the stormwater drainage system. The coastal water level range during the 5-day simulation period was -0.4 to 1.1 m NAVD88 in Jamaica Bay (see sect. 2.5, Fig. 8).

The simulation incorporates meteorological forcing from the North American Mesoscale (NAM) WRF model product (Center Environmental Modeling, 2017) and rain forcing from the Multi-Radar/Multi-Sensor system (MRMS). The WRF-NAM model provides east and north winds, atmospheric pressure, relative humidity, air temperature, and short and long wave radiation data on a 12-km spatial grid and 3-hour temporal resolution. The MRMS QPE data provide 1.11 km spatial and 1-hour temporal resolution, combining radar, satellite, and rain gauge observations with bias correction to offer more accurate precipitation estimates than radar-only products (Zhang et al., 2016). Hourly data provide a robust value for simulating the event, but sub-hourly data could have higher intensities that would create more impact. However, the time resolution of the forcing data precludes sub-hourly analysis. Figure 3 show maximum hourly intensity and accumulated rain during Ida for the simulation duration. The maximum hourly rainfall intensity was 70 mm/hour during Ida, which is a 50-year return period rain event (2.76 inch/hour) based on NOAA Precipitation Frequency Data Server (station ID: 30-5803) (National Weather Service)). Observed rainfall intensity was greatest on the western side of Jamaica Bay. Additionally, the maximum accumulated rainfall, over 5 days of simulation reached 160 mm. Figure 4 shows the time series of rainfall to highlight the sharp peak of rainfall during the 5-day simulation and the absence of significant rainfall before and after.

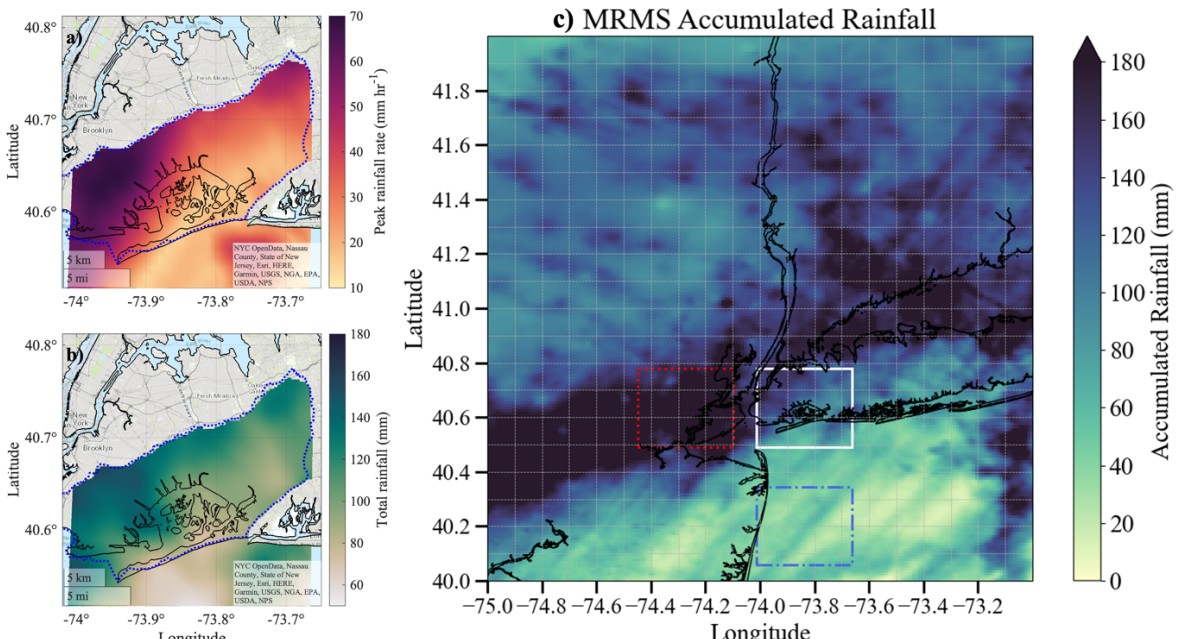

**Figure 3. a) a) Maximum hourly rain intensity for Ida, b) Total accumulated rain during Aug 29, 2021, to Sep 02, 2021, for Ida. (watershed boundary is shown with blue dots, and topographic base maps are from MATLAB, hosted by Esri®), c) Ida total rain is shown as color shading (temporally accumulated rainfall over the 5-days simulation time). The white box shows the Jamaica Bay domain, whereas the red box (dot line) shows the rain that would fall over the domain if the rain were displaced eastward (worst case scenario), and the blue box (dash-dot line) shows what rain would fall over the domain for a displacement northward (best case scenario).**

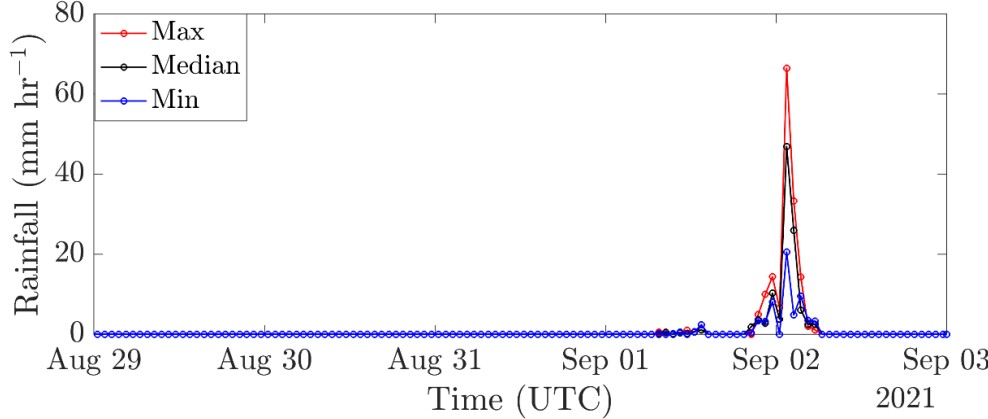

**Figure 4. Rainfall time series at (Max) location with the maximum accumulated rain, (Median) location with the median accumulated rain, and (Min) location with the minimum accumulated rain over the 5-day simulation.**

### 2.4. Drain rate calibration and context

Rainfall into the study region can have several fates, including infiltration, ponding, surface flow, or interception and export through the drainage system. Because the model does not explicitly include infiltration or a stormwater system, we account for the drainage with a collective "drain rate". The drain rate can be influenced by different factors such as the characteristics of the sewer network, land use, soil infiltration rates, and potential blockages in the stormwater system. Given its impact on the results and its relatively unconstrained nature, we start by applying a spatially uniform drain rate and treat it as a calibration parameter. The sensitivity of results to the assumed drain rate is explored with several different values. First, we perform a base simulation with no drain rate with realistic rainfall and atmospheric forcings. This base Ida simulation is initially compared with observations of high-water mark data points, and then different drain rates are investigated to improve the model calibration. Potential drain rate values extend up to 44 mm/hour, which corresponds to the design capacity of the stormwater system (NYC-MOCEJ, 2023).

It is informative to compare our calibrated drain rate with an estimate of the required drainage during Ida, the volume of water that stormwater infrastructure must effectively manage to prevent flooding. The Curve Number (CN) method (Cronshey, 1986), a cornerstone in hydrological modeling, is employed to estimate runoff (the required drainage) from this event. This method is defined by the equation Eq. (1):

$$Q\ (mm) = \left(\frac{(P-0.2S)^2}{P+0.8S}\right) \tag{1}$$

where $Q$ represents the runoff, $P$ denotes total precipitation, and $S$ signifies the potential maximum retention after runoff initiation, calculated based on Eq. (2):

$$S \ (mm) = \left(\frac{25400}{CN}\right) - 254 \tag{2}$$

      To reflect the spatial variability in urban and non-urban areas, a weighted CN of 93.95 is derived based on the land cover categories (C-CAP data, Figure 2d) and soil data from USDA-NRCS Web Soil Survey (U.S. Department

of Agriculture; Cronshey, 1986). We consider the rainfall associated with Ida as a concentrated 3-hour period of precipitation (as vast majority of the rain is over three hours according to MRMS), starting at 02 Sep 2021, 00:30 UTC. A comparison of the gauge data from the CoCoRaHS at Howard Beach station (126 mm) with MRMS data at the same location (102 mm) confirms the reasonable accuracy of MRMS. During this period the rainfall generates an average of 74 mm (2.9 inches) total accumulation across the domain. Using this formula, we determine that Ida would

result in average 58 mm (2.3 inches) of runoff across the domain. This finding, which is based on the rainfall forcing of the event, provides useful context as it denotes the volume of water that the urban stormwater infrastructure must efficiently channel to avert flooding.

### 2.5. Sensitivity analysis

      The sensitivity of flooding in the Jamaica Bay watershed was evaluated using multiple scenarios based on

the realistic forcing by shifting the track and timing of the storm. These hypothetical storm cases are designed to represent worst case and best case precipitation conditions from Ida for the Jamaica Bay watershed.

      At the time Ida passed the New York City metropolitan area, the region with the most intense and widespread rainfall was over New Jersey, 48 km to the west of Jamaica Bay (Fig. 3c). Thus, to examine sensitivity, we shift Ida rainfall and wind field spatially to the east, to have an approximate "worst case scenario". This shift is well-within the

260 potential track uncertainty for Ida, given that the National Hurricane Center (NHC) forecast cone of uncertainty at 12 hours prior to storm passage is 48 km. For comparison, we create a "best case scenario" by shifting Ida northward the same distance, which results in much less precipitation over the Jamaica Bay watershed.

      While Ida's rain set records at Central Park and other locations, much of the Jamaica Bay had far lower rainfall rates. Shifting the storm track east by 48 km increases Ida's rain average across Jamaica Bay watershed over

265 3 hours to 145 mm (5.7 inches), which represents a 3-hour rain event with 500-year return period based on NOAA Precipitation Frequency Data Server (station ID: 30-5803) (National Weather Service), and close to a 100-year storm under climate projections (129.54 mm) based on Northeast Regional Climate Center extreme precipitation projections (Cornell University). During this worst case scenario, all the Jamaica Bay watershed area experiences maximum hourly rain intensity greater than 30 mm/hour, and the watershed area experiencing rainfall intensity greater than 60

270 mm/hour more than doubles. In addition, total rainfall in this scenario over the five days increases by 48% to 237 mm. In the best case scenario, the watershed experiences 60% less total rainfall, equal to 64 mm. Figure 3c shows Ida rainfall over the simulation domain (white rectangle), and the regions that the rainfall came from for the eastward shift (red rectangle, with dot line) and the northward shift (blue rectangle, with dash-dot line).

      We also analyze the timing of Ida with tidal conditions to ascertain whether the synchronization of the peak

rainfall rates with storm surge and high tide could exacerbate flooding. Figure 5 illustrates tidal observation from Bergen Basin (location shown in Fig. 2) throughout the simulation period. The most intense rainfall during Ida

happened on 02 Sep 2021, 01:30 UTC. We consider two alternative temporal scenarios: 1) positioning the maximum intensity of the Ida rain six hours prior to its actual occurrence, corresponding with the earlier high tide, and 2) a delay of nineteen hours to examine the effects of the storm's peak overlapping with the highest tide during the simulation period. The time of the storm's peak rain will be shifted from observed timing (when the water level is 0.3 m NAVD88) to occur when the observed water levels are 1.01 m and 1.15 m NAVD88, respectively.

These experiments are simplistic and true variations and uncertainties in storms can affect a wide range of storm characteristics including intensity and spatial distribution of rainfall. However, a comprehensive study of Ida forecasting uncertainties is beyond the scope of this paper.

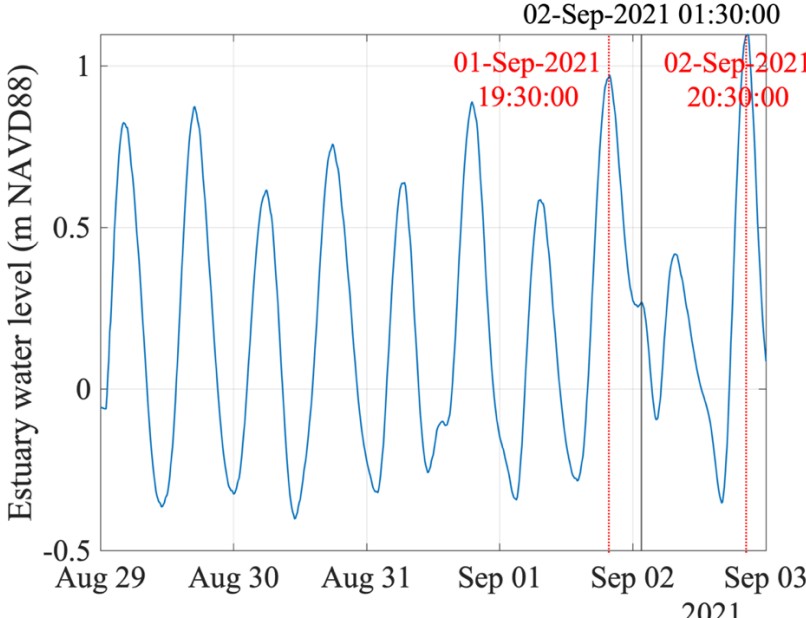

**Figure 5 Tide gauge water elevation at Bergen Basin**

### 2.6. Flood map and building association

This research effort produces the estimated flood depth map of Ida, which is publicly accessible and is utilized by New York City Emergency Management (NYCEM). An interpolated flood map of Ida, based on differencing the modeled peak water elevation and a 30 cm resolution DEM, is generated and published in a separate resource (Kasaei et al., 2024). Also, to quantify the number of buildings affected by Ida and the spatial shift worst-case scenario, a GIS spatial intersect analysis is performed between the model outputs and NYC building footprints in the Jamaica Bay watershed. These are obtained through the NYC Department of Finance Map PLUTO tax lot dataset (DCP).

### 2.7. Limitations

The methods above expand the capabilities of the widely used COAWST model system and enable the first simulation and flood map for Ida. However, several simplifications are worth considering while interpreting our results. The model's simplifications, such as assuming spatially constant drain rates and a bare-Earth land surface, help simplify model setup but do not fully capture the complexity of urban flooding, particularly in areas with variable drainage systems and land use. However, during extreme events like Ida, where stormwater systems were overwhelmed or blocked, these assumptions likely have a reduced impact. Additionally, the moderate ~50-meter

spatial resolution may overlook finer-scale variations in flooding. Potential future improvements to mitigate these limitations are discussed in Sect. 4.

## 3. Results

### 3.1. Model Calibration and Validation

For the baseline simulation (no drainage rate, no spatial or temporal shifting of rain) of Ida's flood impact on Jamaica Bay, the flood depths vary spatially throughout the domain (Fig. 6, discussed below). Flood depths below Dcrit (5 cm) are not tallied in flood depth mapping and calculations. For model validation, we reference surveyed High-Water Marks (HWMs) from USGS website for the event, finding seven locations within our modeled area, one more HWM is from Community Flood Watch Project (for these datasets, see Data Availability section). Furthermore,

to enhance the robustness and generality of the model calibration, we investigate another less-intense rain event that occurred on 29 September 2023. In this event, we identify 10 additional flood gauges from the FloodNet project (Mydlarz et al., 2024). We use the maxima of the flood time series during this event as HWM data points (more information on this rain event in the Supplementary Material), in addition to the HWMs for Ida. Figure 7 presents the model results vs observed HWMs for different drainage rates in both rain events. We investigate metrics of Root Mean

Square Error (RMSE), Mean Absolute Error (MAE), and Nash-Sutcliffe Efficiency (NSE) to assess model accuracy. NSE equals 1.0 for a perfect fit, and 0.0 when the mean of the timeseries is equally predictive as the model (Nash and Sutcliffe, 1970). The analysis shows a tendency for the model to predict higher flood levels, with a Root Mean Square Error (RMSE) of 40 cm and a Mean Error of 13 cm (Fig. 7a). This discrepancy is likely attributed to the model's omission of infiltration and storm water drainage; however, it is important to acknowledge that other factors, such as

uncertainties in atmospheric forcing, could also contribute to this discrepancy. To rectify this, we test different drainage rates in the model and the values that lead to the best results are summarized below. In addition to the HWMs, we investigate the tide gauge data at three stations: Inwood (USGS station 01311850 (U.S. Geological Survey)), Rockaway (USGS 01311875 (U.S. Geological Survey)), and Bergen Basin (a station managed by Stevens Institute of Technology) (location of the gauges are shown in Fig. 6). For the comparison of the calibrated model, we evaluate the

peak water level during the simulation (the first high tide after Ida), where the model has error of +8, +7 and +6 cm for Inwood, Rockaway, and Bergen Basin gauges, respectively (validation timeseries at these three gauges are shown in supplementary material). These tide gauge results are not utilized in the calibration process, as the peak tidal water levels in Jamaica Bay during Ida do not occur during the period of rainfall and were controlled by ocean forcing.

   To simulate the sewer system's capacity within our models, we introduce drain rates of 6 mm/hour (0.25

inch/hour), 13 mm/hour (0.5 inch/hour), and 19 mm/hour (0.75 inch/hour). The 13 mm/hour rate, which provides the best fit to high-water marks from both the main event (Ida) and a less intense event (29 September 2023), reflects a reasonable drainage rate for the model based on the best fit with observed HWMs. Assessing the performance across the three metrics with High Water Marks (HWM) data, we conclude that the 13 mm/hour drain rate is the best overall option. It provides the highest Nash-Sutcliffe Efficiency (NSE) of 0.64, the lowest Root Mean Square Error (RMSE)

of 0.20, and a reasonably low Mean Absolute Error (MAE) of 0.06. While the 6 mm/hour drain rate has a NSE of 0.54 and the lowest MAE (−0.02), its RMSE of 0.23 is slightly higher, and it doesn't outperform the 13 mm/hour rate in overall predictive ability. Since the 13 mm/hour drain rate has the highest NSE, indicating the model's best ability to

explain the variance in observed HWMs, it remains the optimal choice, striking the best balance across all key performance metrics. Figure 7 depicts the improvement of the metrics with applying 13mm/hour (0.5 inch/hour) drainage rate.

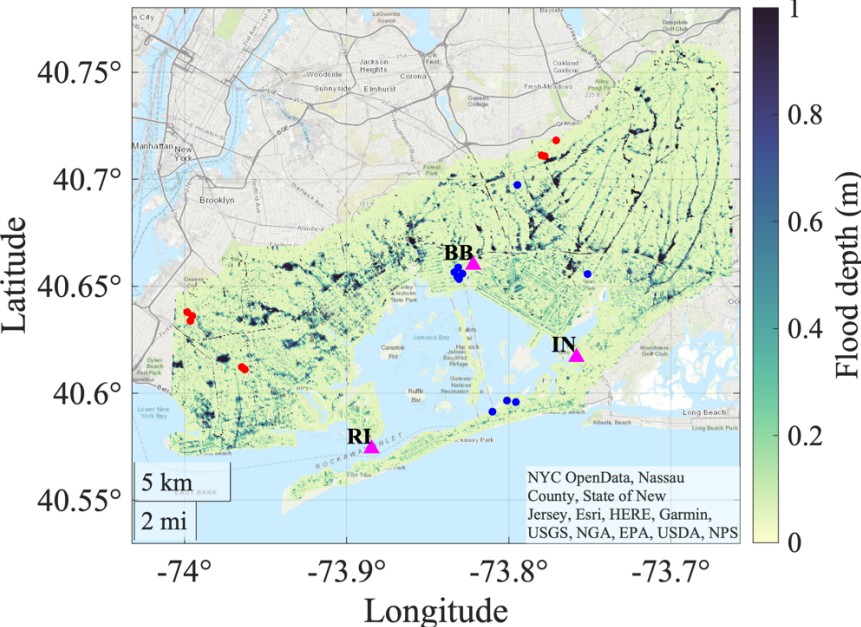

**Figure 6 Flood depth above ground level (model with no drainage), red circles, blue circles and purple triangles show HWMs from Ida, HWMs from September 29th storm, and tide gauge locations, respectively. Tide gage locations Bergen Basin (BB), Rockaway (RI), and Inwood (IN) are shown. (topographic base map from MATLAB, hosted by Esri®)**

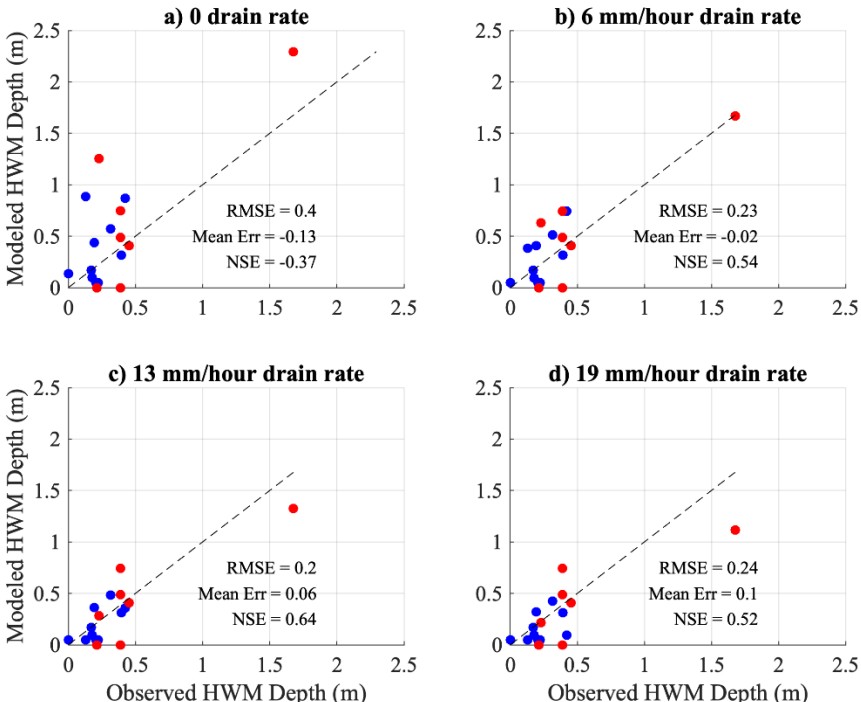

**Figure 7 Model results vs observed HWMs for different drainage rates, used to calibrate model and finalize choice of 13 mm/hour drain rate. Data include HWMs from Ida (in red color) and September 29th storm (in blue color).**

According to the CN calculations in Sect. 2.4, Hurricane Ida is estimated to generate an average of 58 mm of runoff over a 3-hour period, which corresponds to a runoff rate of approximately 19 mm/hour (or 0.75 inch/hour). This establishes the necessary rate at which stormwater must be managed to ensure proper drainage. The model's empirical validation is achieved by comparing simulated flooding depths against observed data, yielding root mean square errors (RMSE) of 23, 20, and 24 cm for drain rates of 6, 13, and 19 mm/hour, respectively. Notably, the 13 mm/hour drain (0.5 inch/hour) rate most accurately reflects observed flooding with the least under/over estimation according to the metrics. Further Ida result maps in this study such as the Ida flood map, flood speed, and the control model to evaluate sensitivity analyses utilize this drainage rate (13 mm/hour). We address the realism of this best-fit drainage rate in the discussion section below. Incorporating drainage into the model improves the results compared to the baseline case when comparing to the high-water marks (HWMs), particularly for the elevated HWMs that are ponding. The HWMs (in Figure 7 plots) that remain unchanged with increasing drainage rates are elevated, steeply sloping locations where water rapidly flows and does not accumulate.

### 3.2. Model results

The simulations provide estimates of peak water depths across the watershed, showing water depths greater than 1 m are generally aligned along major streets or in isolated confluence areas (Fig. 8). For this study, we created general categories of flooding in addition to the maps of depth and extent of the flooding. Flooding up to 0.3 m ("shallow") typically causes minor inconveniences, such as water accumulation and disruptions to pedestrian and vehicular traffic. Floodwaters ranging from 0.3 to 0.9 m ("deep") can interfere with vehicle operation and damage

structures, causing moderate damage and inconvenience. In instances where water depths surpass 0.9 m ("extreme"), there is a substantial risk to both individuals and property, with potential outcomes including the movement of vehicles, mandatory evacuations, and extensive property damage. In the model results, 23% of the urban area in the watershed experiences water depths greater than 0.05 m (Fig. 8). Within this urban flooded area (water depth > 0.05 m), 18% is categorized as shallow flood (88 km$^2$), 4% is deep flood (20 km$^2$) and 0.7% is extreme flood (3.6 km$^2$). About 3% of the urban area (14.5 km$^2$) experiences more than 10 hours of severe flooding (flood depth > 0.3 m) and 9.4 km$^2$ are flooded for more than 20 hours. The severely flooded areas include major streets and low-lying areas. Of the 4621 buildings affected by flooding exceeding 0.3 meters, 21% (970 buildings) experience flooding greater than 0.9 meters.

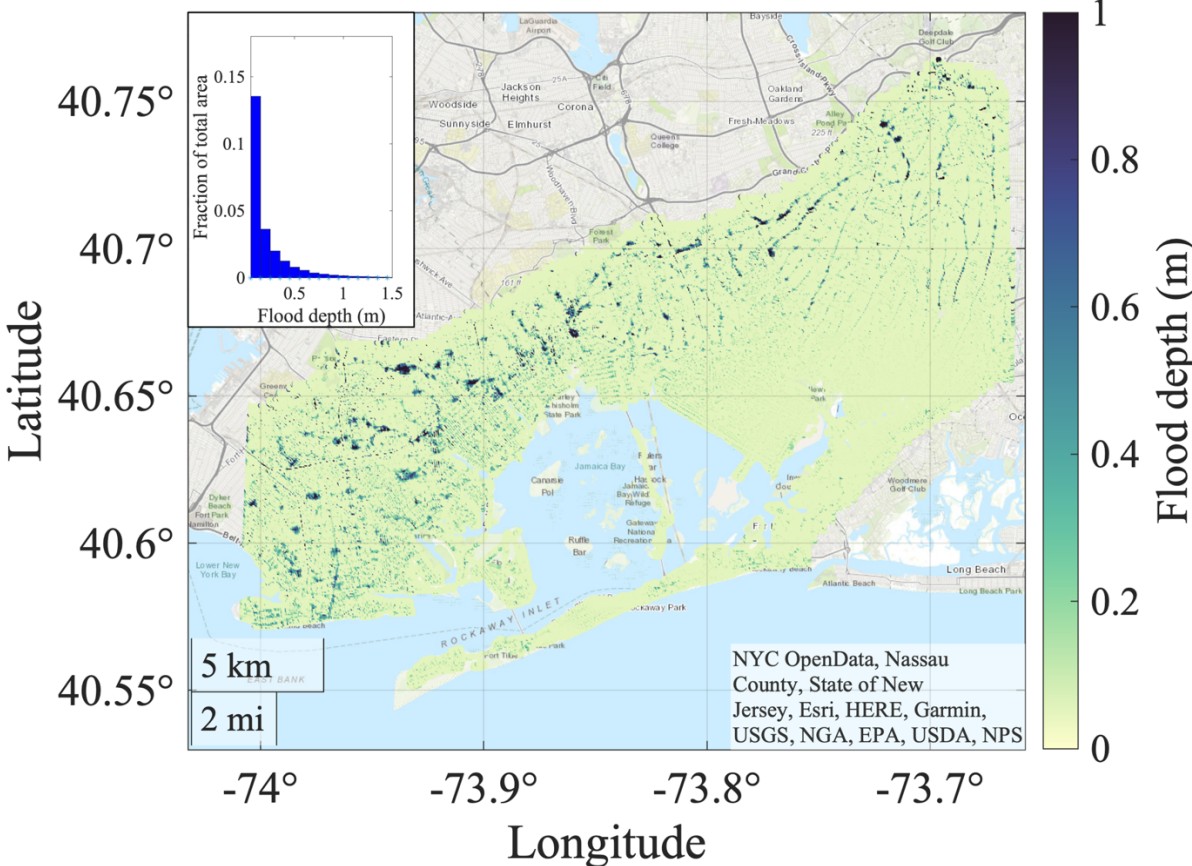

**Figure 8. Map of Ida modeled flood depths, and flood histogram (topographic base map from MATLAB, hosted by Esri®)**

The model results allow for diagnosis of maximum water velocities in flooded areas. The highest speeds, often greater than 1 m/s, occurred primarily in steep areas and along streets (Fig. 9). High speed zones (>1 m/s) affect 0.56% of the urbanized domain, equivalent to approximately 2.7 km$^2$, while moderate water velocities (0.5-1 m/s) account for 1.75% or roughly 8.5 km$^2$ and lower speed regions (<0.5 m/s) constitute the remaining 97.7%, which translates to approximately 472 km$^2$. The maximum flood speed across the domain during Ida reached 3.6 m/s. These estimates of water speed can help identify the potential zones for delineating flood risk and informing mitigation strategies. High-speed areas pose substantial structural and safety challenges, moderate velocities require caution for potential impacts on transport and infrastructure, and low velocities primarily pose concerns related to static

inundation of structures and streets. However, it is important to acknowledge that our velocity data has limitations, as it does not account for buildings, cars or other roughness elements.

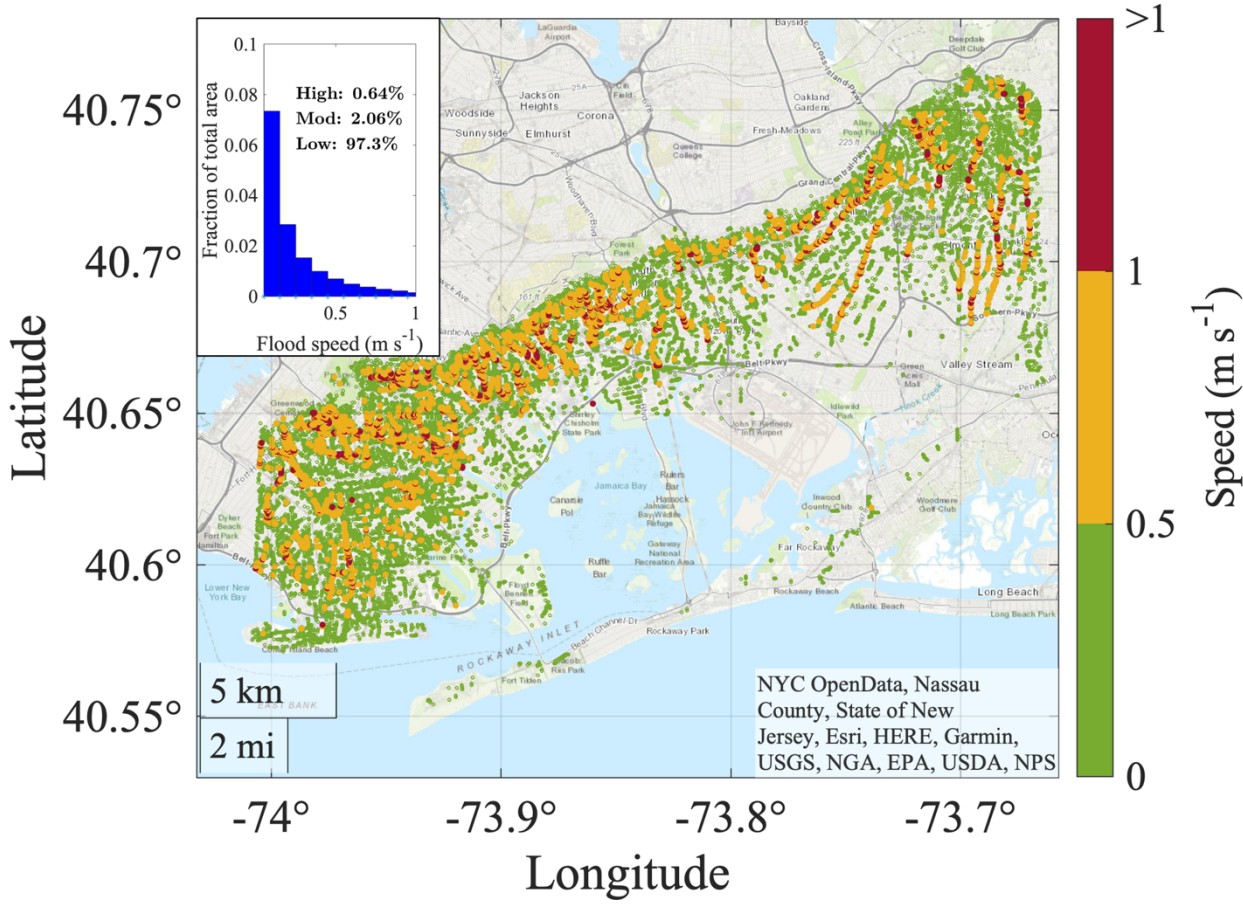

Figure 9 Flood speed greater than 0.05 m/s, and flood speed histogram (High, Mod, and Low stands for High speed, Moderate speed, and Low speed, respectively). (topographic base map from MATLAB, hosted by Esri®)

### 3.3. Sensitivity analysis results

#### 3.3.1. Spatial shifting of the storm

The sensitivity testing demonstrates that large differences in flooding result from relatively small changes in storm track. Shifting the rainband eastward shows that the flooded area could increase by 62%, from 112 km² to 181 km² in total. Moreover, the areas classified as extreme flood (> 0.9 m) and deep flood (0.3-0.9 m) increase by 257% (1.9 km²) and 141% (5.9 km²), respectively. Maximum flood depth rises by 70% from 5.6 m to 9.5 m, and the area with high water velocities (>1 m/s) increases by 257% (6.95 km²) (Fig. 10a). In this scenario, the total number of buildings experiencing flooding over 0.3 meters and 0.9 meters increases by 28% and 46%, reaching 5907 and 1412 buildings, respectively. Adjusting the rainband northward by 48 km and simulating the best case scenario, there is a 73% reduction in predicted urban flooded area, but importantly, this does not eliminate the flooding in the watershed. Although the flood map result shows a reduction in maximum flood depth in this scenario (from 5.6 m to 3.8 m), we still see 30 km² of the urban watershed classified as shallow flood (Fig. 10b). In addition, further investigation of the whole region (not just the urban area) shows there are some areas that are always flooded, such as marshes and the

edges of the bay. Comparison between the worst case scenario, Ida rainfall, and best case scenario shows that as the storm gets milder, the flooded area becomes shallower with significantly less deep and extreme flooding.

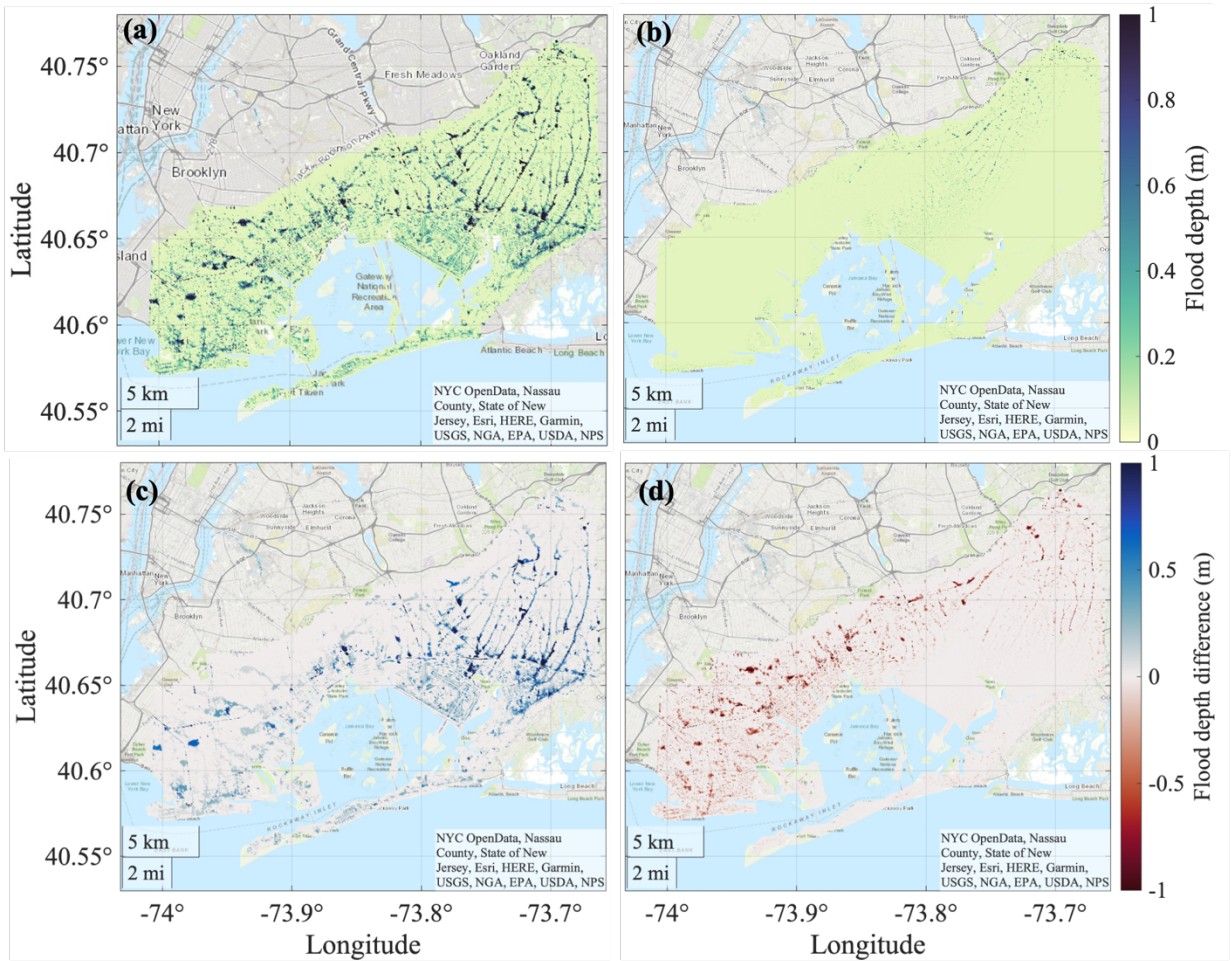

**Figure 10 Flood map for sensitivity analysis for rain shifted eastward (a), and northward (b), as well as flood depth difference in comparison with Ida (flood depth minus control, shown in Fig. 11) for shifted rainband eastward (c), and**
**northward (d). (topographic base map from MATLAB, hosted by Esri®)**

### 3.3.2. Temporal shift of the storm

    Model scenarios with temporal shifts in the storm are examined to investigate the potential amplification of flood severity due to compounding of rainfall, storm surge, and tides. The results indicate that the spatial extent of the inundation mostly remains the same as in the baseline simulation (Ida). However, looking more closely at the coastal
flood plain, and comparing the water level change from Ida and the two temporal shift scenarios (as explained in Sect. 2.5.) reveals evidence of compounding in the second scenario but not the first. When the peak of Ida's rainfall coincides with the storm surge and the highest tide of the 24-hour period, the urban flooded area increases by 1.3% (approximately 1.4 km²). Of this increase, 20% has deeply flooded areas. This additional flooding predominantly affects coastal areas such as the east and west sides of Jamaica Bay, and Hamilton Beach (Fig. 11). In Hamilton Beach,
the second temporal shift scenario (Fig. 11b), results in flood depths reaching 30-40 cm within the neighborhood.

Figure 11b displays model grid cells where the additional flood depth with the temporal shift surpasses 5 cm. These findings underscore the model's utility in representing compound flooding events, such as for hurricanes that bring both extreme rainfall and storm surge (Chen et al., 2024).

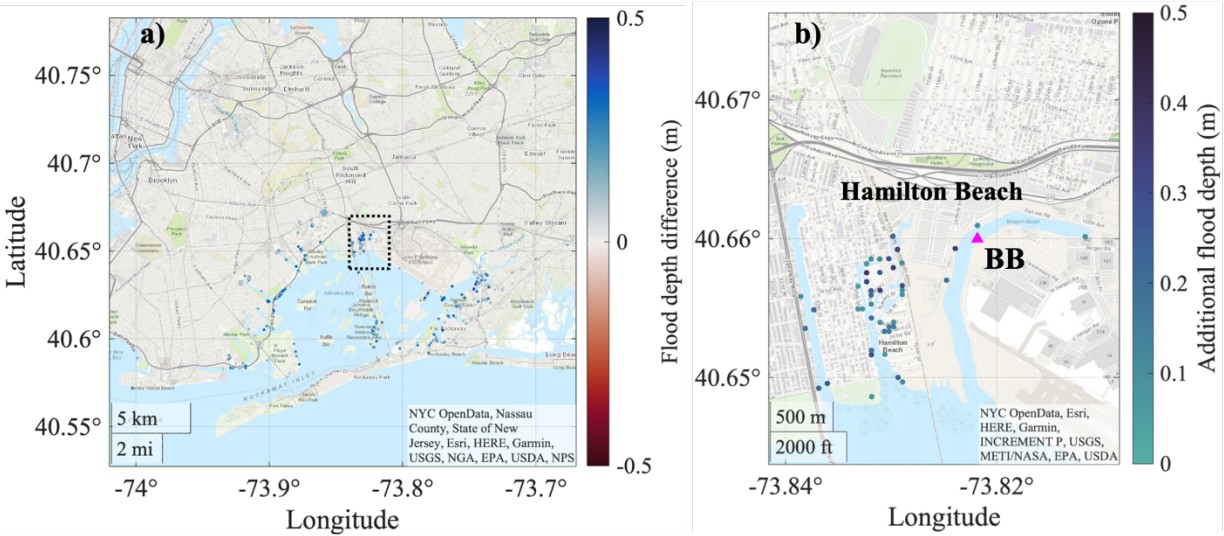

**Figure 11 a) Flood depth difference map for when the timing of Ida's rainfall is aligned to match the highest tide of the 24-hour period (temporal shift scenario2) in compared with Ida (differences exceeding 5 cm are highlighted), b)model grid cells that exhibit additional flood depth (more than 5 cm) around the Hamilton Beach neighborhood (dashed black rectangle in panel a) due to compounding. The magenta triangle 'BB' marks the Bergen Basin tide gauge locations. (topographic base map from MATLAB, hosted by Esri®)**

## 4. Discussion

This study creates the first spatially continuous flood map of Ida, which can help identify and understand Ida's effects far beyond the scattered high water mark data. While static or bathtub mapping can in some cases be effective for coastal floods (New York, 2015), the bathtub mapping effort for the pluvial flooding of Ida was limited to areas within only 250 m of high water marks rather than a spatially continuous map (Capurso et al., 2023). The absence of flood maps for events like Ida highlights that accurate pluvial flood modeling and mapping technologies are not yet broadly available. As a result, our modeling, despite its simplifications, holds significant potential to inform emergency management and mitigation efforts.

To fully comprehend urban pluvial flood risk, it is important to understand how every part of a city responds to extreme rainfall, yet only certain areas of NYC experienced rainfall rates above 50 mm/hour during Ida. The spatial shifts (Fig. 3) exploring both the worst case scenario and the best case scenario depict the range of flood hazard in the Jamaica Bay watershed. Our findings indicate that even a small perturbation in Ida's storm track would have caused even more severe flooding, affecting both the extent and depth of inundation. Furthermore, during the best case storm scenario, the area still experiences widespread shallow flooding.

Additionally, understanding whether, where, and how pluvial and coastal flood drivers interact is vital to assessing a coastal city's flood risk. Compound (pluvial-coastal) flooding remains poorly understood, including such aspects as whether compounding causes an increase in flood area, an increase in flood depth, or both. The temporal

shifting of the storm to coincide the most intense rainfall and storm surge with the high tide shows greater depths of flooding of coastal neighborhoods and a greater extent of inundation (increase by 1.3 km$^2$). This indicates that compound flooding can cause greater danger in coastal flood zones during an extreme pluvial event. The greater footprint of compound flooding could negatively influence emergency management activities (e.g. by blocking roadways in pluvially affected flood areas). These results illustrate the capability of the COAWST to capture compound flood effects, and its utility for future modeling of a wider range of coastal-pluvial forcings to improve our understanding of coastal-urban flood hazard.

Our model's ability to represent spatial variations of flood depth with improved accuracy (Fig. 8) when using a 13 mm/hour drain rate, relative to the simulation with no drainage (Fig. 6), underscores the importance of integrating drainage considerations into urban hydrodynamic simulations. Detailed hydrologic and hydraulic models that aim to fully represent urban stormwater systems and coastal water level boundary conditions are under development. However, the data required to build and validate these models are often unavailable, and such models are rarely applied (Rosenzweig et al., 2021). In their place, new simplified models that represent the Earth system are being utilized for city-scale (Sebastian et al., 2021) and national-scale compound flood modeling (Bates et al., 2021). The approach of calibrating the model to a constant drain rate used in the present study can be of use in these simplified modeling efforts.

According to the curve number method (Eq. 1), the runoff rate of Ida which needs to be managed is 19 mm/hour. The calibrated drain rate of 13 mm/hour from our model is less than the calculated runoff rate. Since our model and the curve number method are based on hourly rainfall data, we can compare them with each other, and this comparison highlights the limitations in the storm water system as the calculated runoff rate exceeds the model's calibrated drain rate. However, we should consider that it is likely that brief, intense rain bursts, rather than the hourly MRMS rain rates that were used in this study, caused most of the flooding by overwhelming the stormwater system. This could be a reason that the calibrated drainage rate in our model (13 mm/hour) is less than the current design storm water system capacity, 44 mm/hour, although certain areas do not have modernized storm sewer infrastructure (NYC-MOCEJ, 2023). Also, stormwater systems often become blocked in heavy rain events, due to garbage or vegetation. As a result, our modeling approach may be more applicable for extreme rain events, though it likely also has a declining accuracy, or need for new tuning, for weaker rain events. Our model results suggest that the deepest flooding occurs in major streets, turning them into river-like pathways. However, the assumption of spatially uniform drain rates may lead to misestimation of the spatial distribution of flood depths. Although major streets may possess better stormwater infrastructure than minor streets, our final high-water mark (HWM) results indicate that the model maintains good accuracy.

The improved COAWST model has the potential to more comprehensively simulate the coastal system and factors contributing to flooding, relative to typical urban hydraulic-hydrologic models. The result is a coastal system model capable of simulating tides and storm surge, rainfall, wind wave overtopping, erosion and air-sea-wave interactions. Research made possible by this new model includes infrastructure adaptation planning for urban coastal pluvial flood studies, analyses of rain influence on estuary hydrodynamics, and has the potential for future studies on

coupled pluvial-coastal flood-induced sediment transport and erosion. This capability is not readily available in many existing models, which often require separate or one-way coupled models to achieve similar results.

The model also has the potential for improving flood hazard and adaptation assessment. Many recent studies have assessed joint probability of rain and surge for an urban environment (e.g. Kim et al., 2022; Zellou and Rahali, 2019), yet very few have modeled flooding for a range of these scenarios. Our sensitivity analyses, incorporating the potential variability in storm tracks and storm timing, demonstrate the promise of the model to capture a wide range of flood forcing scenarios and show the importance of storm track variability and timing for flooding.

Despite these advancements, this study is not without limitations which are mentioned in Sect. 2.7. The model simplifications could be modified in future work to capture a more complex and thorough evaluation of urban flooding. For example, future work could experiment with applying spatially varied drain rates based on storm water system data and land use type data. The empirical validation could also benefit from more extensive HWMs or spatial maps of observed flooding to more fully evaluate the model's reliability. These future improvements are possible but large

undertakings, as the detailed information on storm water pipe systems are not easy to gather and include, and the data for validation from this storm are limited. However, a concurrent rapid increase in urban flood observations (Gold et al., 2023; Mydlarz et al., 2024) could be a remedy for this data shortage, which could help for future flood events and more detailed model development. Our research retrospectively analyzes and models Ida using the best estimate of rain from MRMS. However, uncertainties in precipitation products can affect the accuracy of our resulting flood

simulation, especially at fine temporal and spatial resolutions. Previous studies (e.g., Xu et al., 2025) have shown that precipitation data uncertainties, stemming from differences in spatial resolution and input sources, can lead to substantial variations in simulated flood extent. Additionally, Feng et al. (2024) demonstrated that using multiple atmospheric forcing datasets helps to better capture the uncertainty in hydrological simulations. Future research could explore the sensitivity of our findings to different precipitation datasets, following similar ensemble-based approaches.

**5.    Summary and Conclusions**

In this research, a coastal system model (COAWST) is enhanced to capture the volumetric effect of rainfall in the ocean and on floodplains. A simplified drain rate capability is added to account for stormwater system and infiltration effects on flooding. These improvements are applied in a simulation of flooding by post-tropical cyclone Ida in the Jamaica Bay watershed of New York City. The calibration and resulting accuracy of the model compared

with empirical High-Water Marks (RMS error 20 cm) and maximum water levels in the estuary illustrate the model's predictive capabilities yet suggests a need for improvements in modeling detail or sophistication for capturing a wider range of rain intensity events.

Outcomes of the research include a spatially continuous flood map of Ida, and improvements to our understanding of the flood hazards posed by extreme rain events, as well as a developed model capable of investigating

compound (pluvial-coastal) events for the Jamaica Bay area. The sensitivity analysis depicts the flood hazard associated with changes in storm track and timing for Jamaica Bay. It reveals that flooding from Ida could have been worse due to shifts in the location of the most intense rain or co-occurrence with storm surge and high tides. Capturing the compounding effect is particularly important, given that compound floods are expected to become more common and important as sea levels continue to rise (e.g. Mita et al., 2023). Effective translation of this scientific knowledge

into coastal and urban planning is imminent, requiring interdisciplinary efforts and long-term studies of the risks of future climate challenges.

## 6. Competing interests

The authors declare that they have no conflict of interest.

## 7. Acknowledgements


This work was supported by the U.S Geological Survey through the Extending Government Funding and Delivering Emergency Assistance Act (Public Law 117-43, award G22AC00399-00) under the North Atlantic Coast Cooperative Ecosystems Study Unit (NAC-CESU). Any use of trade, firm, or product names is for descriptive purposes only and does not imply endorsement by the U.S. Government.

We would like to acknowledge the NYC Emergency Management GIS team who helped generate high resolution Ida flood map and building exposure numbers.

## 8. Dataset Availability

The improvements to COAWST are embedded in v3.8, which is available for download at: https://code.usgs.gov/coawstmodel/COAWST. The model setup files, flood depth dataset, Bergen Basin tide gauge

dataset is published in Mendeley (Kasaei and Orton, 2025). Final flood map for Ida is published in Mendeley ((Kasaei et al., 2024); a collaboration with NYC Emergency Management). Meteorological forcing for the Jamaica Bay model was provided by the WRF model, North American Mesoscale (NAM) product, the data can be accessed: https://www.ncei.noaa.gov/thredds/catalog/model/model.html. USGS High Water Mark data points can be accessed: https://stn.wim.usgs.gov/FEV/#2021Ida. Community Flood Watch HWM data point can be accessed:

https://mycoast.org/search-reports?state=ny&fwp_categories=highwater%20. Tide gauge data from USGS can be accessed: https://maps.waterdata.usgs.gov/mapper/index.html.

## 9. Author contribution

The paper and the experiments were conceptualized by SK, PMO and JCW; the parent (regional model) simulation was performed by DKR, and Jamaica Bay simulations by SK. The COAWST model improvements were

done by JCW; statistical analyses by SK with help from PMO. The original draft was written by SK with help from PMO; further review done by PMO, DKR, and JCW, and the edits done by SK with help from PMO. Project administration was performed by JCW and PMO, and funding acquisition was done by JCW and PMO.

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
