# Peer review of "Pluvial and potential compound flooding in a coupled coastal modeling framework: New York City during post-tropical cyclone Ida (2021)"

_EGUsphere, 2024_

## Referee Comment (RC1)

**Summary**

The manuscript describes improvements made to an existing hydrodynamic model, COAWST, that will now include a uniform drainage rate as a volumetric source to represent the urban drainage system and infiltration. The improved model was tested over the Jamaica Bay watershed (NYC) during Hurricane Ida's impacts. Results are compared with limited high-water marks, and a sensitivity analysis was performed to see the system performance under the variation of the storm impacts.

**General Comments**

The research work proposed by the authors is interesting and might be suitable for the selected journal. However, the manuscript needs substantial changes to reach the expected standards in top-tier journals like this one. Thus, if my comments are addressed, maybe the manuscript will be more suitable for publication.

First, there seems to be a misunderstanding of the terminology and hydrologic concepts throughout the manuscript. For example, the title states pluvial and compound flooding, but compound flooding may already include pluvial drivers. I suggest the authors follow a nomenclature of pluvial, fluvial, and coastal flood drivers and specify in their scope that they will be considering only the pluvial-coastal interaction. Thus, they can reference this as a compound flood. See line 65 for another example of a pluvial compound. Please revise all the terms used in the paper.

Second, the manuscript format is not suitable. For example, when researchers introduce a new model or modification, they should add a limitation section before the results so the reader is aware of this upfront. However, despite the study's many limitations, the authors only summarize in the last paragraph of the discussion. I think the authors have more than enough to have a robust limitation section. Also, the introduction lacks a clear research gap/motivation for the work. So far, I can interpret from the introduction the importance of flood modeling but not what others have done regarding pluvial-coastal flood modeling in urban settings and the rationale for making these improvements to the COASWT model. Thus, I urge the authors to give some context to the previous works in this field, present the knowledge gap, and discuss their research questions that will bridge this gap. Also, Lines 82-88 disrupt the story-telling flow that all peer-reviewed journals should have, especially in the introduction. Thus, please rephrase. Furthermore, the description and figures of Ida's event should be moved to section 2.1. In the results section, the authors repeat too much or/and give too many details when describing the results.

Lastly, my biggest concern with the manuscript is the substantial simplification of the hydrologic process the authors did for the model improvement. I can support a simple

approach, but the authors need to then justify their selection. Furthermore, the authors have many vague statements in the discussion and conclusion section that are not properly supported by the results and/or the model simplification. For example, I would not suggest stakeholders and decision-makers use the results of this model to provide flood-resilience measures to avoid Ida's flood impact since the authors simplified the storm drainage system and infiltration as a uniform drainage rate over the entire basin on the model. Also, their rainfall addition is not included in the governing equations as a source code and only as a volumetric addition, which can produce different hydrodynamic behaviors expressed by previous studies. At a bare minimum, the authors should compare their approach with other rain-on-grid models. I think the authors need to defend and justify their selected approach with more detail.

**Specific Comments**

- L12: replace all with most, remove highly, and replace vulnerable with prone.
- L19: RMS is not define before using its acronym.
- L32: remove "and motivation" from the section header.
- L49-51: the statement needs support from a reference.
- L61: how about also the overestimation? Studies have shown that not accounting for all the physical processes correctly can result in both under and overestimation.
- L71: ICPR model is now called StormWise, so please include this name for future reference of the readers.
- L73-74: Is it crucial for compound flood simulations to have 3D hydrodynamics? The authors claim this as crucial and even one of their novelty, but they fail to provide evidence of such need in the field of the compound flood. Please justify why this is needed with cited literature.
- Figure 1: The watershed is barely noticed on panel A, so increase the line width. The authors can use the USGS HUC watershed shapefiles for this. Also, I am almost certain that the authors do not have copyright permission to include a figure from Wikipedia as the one in panel b, so please make your own figure. When you make your own figure, please describe what the color points mean. Lastly, this figure should be moved to Section 2.1.
- Section 2.1: the authors should include a brief summary of other storms that affected the watershed and its response to the system, such as Hurricane Sandy.
- L126-128: does this mean that the rainfall component is not directly integrated into the governing equation as a source term? Please explain better and justify your approach. Several authors have included rainfall on coastal models by modifying their governing equations, like Dresback et al. (2022) and Santiago-Collazo et al. 2024).
- L129-132: the authors should comment more about the limitations their selected approach for drainage rate affects accuracy and real-life scenarios. For example, the spatial variation of the stormwater infrastructure, the temporal-varying drainage

rate of the system during the event, and the backwater flow preventer structures placed typically in the sewer outlets, to mention just a few.

- L132-133: I disagree with the author's statement that the model does not need to route the remaining runoff toward the ocean for their study. However, they claim the importance of their model for compound flood assessment, but the first point where you exhibit compound flood in coastal urban cities is the drainage outfall and how it propagates inland through the system. Obviously, the authors' proposed model is far from capturing this, but they still claim this (L363-364 ). I will be careful with their statements, especially if they cannot properly back up their statement with results.
- L150: the authors should give more details about the hot start simulation they performed to initialize the model.
- L156-157: the authors should locate these gauges on a Figure. The reader should not wait until the results to find the gauge. The same occurs with other gauges mentioned in the methods section.
- L158: if you are using coastal gauge water levels to represent storm surge conditions, then you are not modeling storm surge directly on your model with a wind and barometric field pressure. This isn't very clear to me and will not be clear to the reader. Please rephrase. Also, if the answer to my question is true, the authors are not modeling storm surge directly; rather, they are just propagating a coastal flood inland. This will need clarification throughout the entire manuscript to change what you call "simulating storm surge" to "simulating coastal flood." The reader will expect that if you are simulating storm surge, then you are applying wind stress to the ocean domain in your model and not just a water level at the boundary condition.
- L155-160: Authors should add a figure of the model domain with a clear description of the model boundary conditions. This will add the explanation in this paragraph.
- Figures 2-3: combine them into a single figure with two subpanels. There are many similar instances of this, such as Figures 4-5. Please try to achieve this since the fewer figures, the better for the reader.
- L170-174: would this mean that you only simulated pluvial flooding and not compound flooding since your coastal flood was minor during this event? If true, the authors should be more clear on their terminology and manuscript language. The reader will expect a "good amount" of both flood drivers to call the event a compound flood. Please be more clear with this earlier in the manuscript. I want to say that the authors should have selected a different event, such as Hurricane Sandy since Ida did not bring significant coastal floods. Furthermore, the justification of selecting Ida as the event due to data for calibration is not supported since you cannot accurately calibrate a whole watershed with only a handful of high-water marks, especially if they are in places where water does not accumulate greatly. The reader will question why you did not model Sandy, so please include a justification for this and why Ida was best. If your purpose was to simulate extreme rainfall, Ida is fine, but if you wanted to simulate a compound flood event, then Ida was not ideal.

- L173: quantify the adjective "far below". Similarly, the authors have many vague adjectives that need quantification, such as "significant," "accurate," and "good." What for the authors is far below could be different to me.
- L180-186: I consider this too much detail about the radar rainfall source. The authors should use only one sentence and cite a reference unless they extensively use rainfall for their model, which is not the case, I believe.
- L189: how does this 70 mm/hr of rainfall compare with other events at the basin? For example, the authors could compare this with return period values from the NOAA Atlas 14.
- L192-193: remove this sentence.
- Figures 4-5: include the basin boundary on the map so we can assess visually the amount of rainfall that falls inside.
- Figure 6: Why show six days of rain if everything happens on a single day? I consider even the graph unnecessary since it gives the impression that the authors are simulating uniform rainfall using the values on this graph.
- L212: what do the selected values of drain rate physically mean for the system? Do they have any physical justification for being selected, or are they just random values that work?
- L221: all of the urban areas the same? The authors are using a typical value of CN for their whole area. However, areas do change, and the soil type and antecedent conditions also play a role in the curve number value selected. The authors should talk more about this since it is crucial for their results, and not just say everyone uses 90 and that it. I strongly recommend the authors compute their own weighted average CN for their watershed using all the parameters needed since that would be more defendable than the current approach. In my opinion, the authors have been too simple on the hydrologic side of the study, which is crucial for any compound flood simulation.
- L222: Can the authors confirm the rainfall amount in 3-hr, using rain gauges within the basin instead of only depending on the radar data, which will not always be 100% accurate? For example, authors could use the CoCoRHAS network.
- L225-226: this is another big limitation that is not addressed in the manuscript. There are many models out there that use the spatially-varying rainfall to generate spatially-varying runoff instead of just a single runoff volume for the entire basin. For example, how would the results change if the authors used the average rainfall over that basin instead of the maximum value? Those are key questions that need to be answered within the sensitivity analysis to justify the approach.
- L226-227: While I agree with the authors' statement, it is not appropriate for their study due to the gross simplification of uniform rainfall and drainage rate. Designing stormwater infrastructure requires more than this since as you keep going downstream towards the outlet, your main sewer line will start receiving more runoff than upstream or other collecting pipes with a smaller catchment area.
- L234: if you only shifted the rainfall field, means that you did not shifted the wind field. Thus, you compound simulation is not well represented. Futhermore, this

statement confirms for me that you did not simulate storm surge over the domain as my previous comment suggests.

- L238-240: this information should be in section 2.1. Also, why not use the return period values at NYC and use the Northeaster states? A local comparison could benefit since a 10-yr return period is not extreme.
- Figure 7: combine with Figures 4-5.
- L257: why not synchronize the peak storm surge with the high tides also? You already had low storm tide conditions. The suggestion made by the authors would not affect your compound flood since the storm tide conditions were already low. This can be shown in the results. If the author manipulates the water level boundary conditions to coincide the peak storm surge with high tide, then they could have a more significant compound flood in the model than what they have now.
- L268: are you referring to the "no infiltration" to the "drainage rate" component of the research? The authors never computed infiltration in the method section. I know that, in theory, it is the same sink term, but this could confuse the readers, so be consistent with your terminology.
- L269: it is technically impossible to see the flood depths to vary spatially since you apply uniform rainfall, drainage rate, and runoff. What you see is a deeper flow in low-lying areas since it tends to accumulate, but a varying rainfall, drainage rate, and runoff (which is more realistic) will show varying floods.
- L270: what do the authors mean by Derit?
- L271: why do the authors label the high-water marks as empirical? Are they were not field surveyed? Explain more.
- L273-275: remove the sentence since this is obvious when comparing data to high-water marks.
- L275: the location of the high-water marks with respect to the deeper flood locations is another limitation of the study that the authors should highlight in the corresponding section.
- L286: I would not call what the authors did a hydrologic model.
- L287: Why did the authors select these drain rates? Justify and put into context what these values might represent. For example, an X-in pipe is flowing full. That would make it easier to relate to the real world.
- L296: the authors mention here that the Cn calculations were used using hourly rainfall, but in the respective section, they said they use the maximum value of rainfall over the basin within a 3-hour time window. Therefore, these arguments confuse the reader. Please decide what hourly runoff or a single value you used.
- L305: what do the authors mean by "non-zero depth HWMs"?
- L314-317: can you reference someone else who has followed a similar flood classification? Also, why is important to classify them as this? Also, I would expect the authors to change the color scale on the results to use this classification rather than the numeric values. If not, then what is the purpose of the classification in the first place?

- Figure 11: why not include the histogram insert in Figure 9? Also, the authors should combine this figure with Figure 9.
- L361: it is not clear what is the second scenario within the context of the sentence.
- L363-364: the authors do not have the necessary results to support this claim. For example, if the authors have run three different flooding scenarios (rainfall only, storm tide only, and compound flood), then they could comment on how important are each of those flood drivers in their modeling approach and in the compound flood assessment.
- L376-377: I disagree with the authors. While simple models could be useful, the authors fail to show this on the manuscript. Thus, it results does not support the claim.
- L381: the authors use the concepts of vulnerability and risk throughout the entire manuscript. However, they really only focus on the hazard component. To talk about risk the authors need to talk about the socio-demographics and the community's exposure. I will move away from these concepts since it is not the main focus of the study and replace it with the term hazard.
- L388: can this only be because tides? Authors should show results with and without rain and tides to see how each driver interacts and they could make a similar statement.
- L389: the authors talked about coastal flood zones but failed to define them. I will suggest citing Bilskie and Hagen (2018).
- L391-393: the authors do not support this statement with their current results.
- L396-398: There are already models that exist and are similar to the proposed approach but without the simplification taken here. For example, ICPR (known as StormWise) can include stormwater infrastructure, spatially-varying rain, and runoff, and include coastal boundaries similar to the author's approach.
- L404 and 406: the word "run off" should be together like "runoff".
- L423: the authors should not claim the sediment transport and erosion component of the model since it was not tested.

---

## Author Comment (AC1)

The authors thank the reviewers for a very detailed reading of the paper and substantive comments that have clearly improved the research and its presentation. Below are the original comments in blue color and italics and our responses below them.

**Anonymous Referee #1**
*Received: 30 Aug 2024*

**Summary**
*The manuscript describes improvements made to an existing hydrodynamic model, COAWST, that will now include a uniform drainage rate as a volumetric source to represent the urban drainage system and infiltration. The improved model was tested over the Jamaica Bay watershed (NYC) during Hurricane Ida's impacts. Results are compared with limited high-water marks, and a sensitivity analysis was performed to see the system performance under the variation of the storm impacts.*

**General Comments**
*The research work proposed by the authors is interesting and might be suitable for the selected journal. However, the manuscript needs substantial changes to reach the expected standards in top-tier journals like this one. Thus, if my comments are addressed, maybe the manuscript will be more suitable for publication.*

1. *First, there seems to be a misunderstanding of the terminology and hydrologic concepts throughout the manuscript. For example, the title states pluvial and compound flooding, but compound flooding may already include pluvial drivers. I suggest the authors follow a nomenclature of pluvial, fluvial, and coastal flood drivers and specify in their scope that they will be considering only the pluvial-coastal interaction. Thus, they can reference this as a compound flood. See line 65 for another example of a pluvial compound. Please revise all the terms used in the paper.*

   The reviewer is correct in that Ida was not both a "compound" flood and a "pluvial" flood at one time. We agree that one storm cannot be both and that compound flooding can include the pluvial driver. Ida caused pluvial flooding, but we also simulate a counterfactual compound (coastal and pluvial) flooding. We have edited the title to make clear that the compound flood was hypothetical ("potential compound flooding").

2. *Second, the manuscript format is not suitable. For example, when researchers introduce a new model or modification, they should add a limitation section before the results, so the reader is aware of this upfront. However, despite the study's many limitations, the authors only summarize in the last paragraph of the discussion. I think the authors have more than enough to have a robust limitation section.*
   Section '2.7. limitations' is added as reviewers suggested.

3. *Also, the introduction lacks a clear research gap/motivation for the work. So far, I can interpret from the introduction the importance of flood modeling but not what others have done regarding pluvial-coastal flood modeling in urban settings and the rationale for making these improvements to the COASWT model. Thus, I urge the authors to give some context to the previous works in this field, present the knowledge gap, and discuss their research questions that will bridge this gap.*
   Change made. We revised the Introduction, specifically the section related to the research gap and research questions so that it shows more clearly the gap and research questions in this study.

4. *Also, Lines 82-88 disrupt the story-telling flow that all peer-reviewed journals should have, especially in the introduction. Thus, please rephrase.*
   Change made. We have revised the section to keep the story telling flow.

5. *Furthermore, the description and figures of Ida's event should be moved to section 2.1. In the results section, the authors repeat too much or/and give too many details when describing the results.*
Changed made. All the description related to Ida moved to Section 2.1. (see specific comments #8)

6. *Lastly, my biggest concern with the manuscript is the substantial simplification of the hydrologic process the authors did for the model improvement. I can support a simple approach, but the authors need to then justify their selection. Furthermore, the authors have many vague statements in the discussion and conclusion section that are not properly supported by the results and/or the model simplification. For example, I would not suggest stakeholders and decision-makers use the results of this model to provide flood- resilience measures to avoid Ida's flood impact since the authors simplified the storm drainage system and infiltration as a uniform drainage rate over the entire basin on the model. Also, their rainfall addition is not included in the governing equations as a source code and only as a volumetric addition, which can produce different hydrodynamic behaviors expressed by previous studies. At a bare minimum, the authors should compare their approach with other rain-on-grid models. I think the authors need to defend and justify their selected approach with more detail.*

We acknowledge the reviewer's concern about the simplification of the hydrologic process in our model, particularly the use of a uniform drainage rate across the basin. As noted in our original manuscript, this modeling approach was chosen because of the difficulty of obtaining detailed and reliable stormwater infrastructure data. In the case of New York City, these data are not public due to security concerns and would need to be obtained from multiple government entities (e.g. New York City and the Port Authority of NY/NJ). However, our experience with the model calibration for two different rainfall events has shown that under intense rainfall conditions, where the stormwater system is overwhelmed, the modeling can nevertheless have acceptable accuracy. This is evident from the high-water mark (HWM) observations (now at 18 locations), which agree well with our model outputs. We have added details of this more detailed calibration to the revised manuscript and supplementary material for further clarity.

Furthermore, while the current simulation of Ida uses a uniform drainage rate, the model itself is fully capable of incorporating spatially variable drainage rates. In the original manuscript this was discussed in the "Future Improvements" section of the discussion, where we outlined plans to update the model once more detailed stormwater system data becomes available. Therefore, while this simplification was a limitation for this simulation of Ida, it is not an inherent limitation of the modeling approach itself.

Regarding stakeholder use of the model, we would like to emphasize that the current model has already resulted in data utilized by NYC Emergency Management for understanding Ida and the 2023 event described in the Supplementary Material. They find it useful for estimating where flooding occurred and where there may be vulnerability to flooding, as they have no other source of Ida flood map data. Our collaboration has resulted in the publication of a Version 1 flood map for Hurricane Ida (Kasaei et al., 2024). This provides evidence that the model, even with its current limitations, is of value to stakeholders. We have also addressed and clarified this point in the text (please see Comment #48 for further details). In this case, the managers understand the simplifications of the modeling approach and are not making detailed infrastructure decisions based on the results.

Concerning the rain addition, you are correct that the original manuscript wasn't clear about how the volumetric addition of water was incorporated into the model. We have clarified the text to say that the rainfall is incorporated into the governing equations of the model. We have provided additional details on this in response to Comment #10 to clarify how rainfall is handled within the model.

Lastly, we have revised several statements in the discussion and conclusion sections (see Comments #47, 48, and 52) to eliminate any vague language and ensure that our findings and interpretations are directly supported by the results.

We hope these clarifications address your concerns.

**Specific Comments**

1. *L12: replace all with most, remove highly, and replace vulnerable with prone.*
Change made.

2. *L19: RMS is not defined before using its acronym.*
Change made to read 'Root Mean Square Error (RMSE)' .

3. *L32: remove "and motivation" from the section header.*
Change made.

4. *L49-51: the statement needs support from a reference.*
Change made. Reference added at the end of the sentence. '(Wahl et al., 2015)'

5. *L61: how about also the overestimation? Studies have shown that not accounting for all the physical processes correctly can result in both under and overestimation.*
Change made. We improved the paragraph context and wording.

6. *L71: ICPR model is now called StormWise, so please include this name for future reference of the readers.*
Change made. 'which is now called StormWise' is added.

7. *L73-74: Is it crucial for compound flood simulations to have 3D hydrodynamics? The authors claim this as crucial and even one of their novelty, but they fail to provide evidence of such need in the field of the compound flood. Please justify why this is needed with cited literature.*
Change made.
"COAWST also incorporates three-dimensional hydrodynamics which is important for accurately predicting baroclinic and stratification effects on storm tides in coastal and estuarine areas (Orton et al., 2012; Ye et al., 2020)."

8. *Figure 1: The watershed is barely noticed on panel A, so increase the line width. The authors can use the USGS HUC watershed shapefiles for this. Also, I am almost certain that the authors do not have copyright permission to include a figure from Wikipedia as the one in panel b, so please make your own figure. When you make your own figure, please describe what the color points mean. Lastly, this figure should be moved to Section 2.1.*
Change made. Watershed added, and storm track plotted manually, and the figure moved to section 2.1.

9. *Section 2.1: the authors should include a brief summary of other storms that affected the watershed and its response to the system, such as Hurricane Sandy.*
Change made. This part added at the end: "… the prior major disaster in the region of Hurricane Sandy (2012) was predominantly a coastal flooding event that severely impacted neighborhoods surrounding Jamaica Bay and spurred major efforts focused on strengthening coastal defenses (USACE, 2022)."

10. *L126-128: does this mean that the rainfall component is not directly integrated into the governing equation as a source term? Please explain better and justify your approach. Several authors have included rainfall on coastal models by modifying their governing equations, like Dresback et al. (2022) and Santiago-Collazo et al. 2024).*

Thank you for your comment. We would like to clarify that rainfall is directly integrated into the governing equations in our COAWST-ROMS model. Specifically, we modified the vertical momentum equation to include the vertical displacement of the water level due to the volume of rain, affecting the free surface and water column dynamics. Additionally, the continuity equation was adjusted to account for the rainfall's volumetric contribution, ensuring proper mass balance in the system. This approach aligns with studies like Dresback et al. (2022) and Santiago-Collazo et al. (2024), where rainfall is incorporated by modifying the governing equations. By integrating rain into both momentum and mass balance, our model comprehensively captures its effects on coastal dynamics.

So, we added this clarification to the text: "Rainfall is directly integrated into the governing equations in the model. The vertical momentum equation is modified to include the vertical displacement of the water level due to the volume of rain, affecting the free surface and water column dynamics. Additionally, the continuity equation includes a source term to account for the rainfall's volumetric contribution. This approach aligns with other studies (Dresback et al., 2023; Santiago-Collazo et al., 2024), where rainfall is incorporated by modifying the governing equations."

11. *L129-132: the authors should comment more about the limitations their selected approach for drainage rate affects accuracy and real-life scenarios. For example, the spatial variation of the stormwater infrastructure, the temporal-varying drainage rate of the system during the event, and the backwater flow preventer structures placed typically in the sewer outlets, to mention just a few.*

Change made. This part added to section '2.7. Limitations'.

12. *L132-133: I disagree with the author's statement that the model does not need to route the remaining runoff toward the ocean for their study. However, they claim the importance of their model for compound flood assessment, but the first point where you exhibit compound flood in coastal urban cities is the drainage outfall and how it propagates inland through the system. Obviously, the authors' proposed model is far from capturing this, but they still claim this (L363-364). I will be careful with their statements, especially if they cannot properly back up their statement with results.*

The reviewer is correct that the model will not capture processes involving actual routing of the flood waters through the stormwater system. We note the paper already included a caveat that … [model simplifications such as spatially uniform drain rates could be modified in future work to capture a more complex and thorough evaluation of urban flooding (line 431-432 of the original manuscript)].

The statement on line 363-364 was "These findings underscore the model's utility in representing compound flooding events, such as for hurricanes that bring both extreme rainfall and storm surge." As demonstrated by the observations and our modeling of Ida, the stormwater system did not function properly during this extreme event. Whether or not the stormwater pipes and blockages of outfalls due to high sea levels are represented in the model is of only secondary importance in such flood events.

As a result, we do feel that the model is useful for studying compound extreme events and we have kept that text.

Appreciating the reviewer's point, we have added a limitations section, and it includes the suggestion for future work could include full coupling with a hydraulic stormwater system model.

*13. L150: the authors should give more details about the hot start simulation they performed to initialize the model.*

Change made. More details were added:

"The Ida simulation commences from a state of rest and temperature and salinity are initialized as spatially constant values, although the full 3-D salt and temperature fields could be utilized to capture baroclinicity and stratification effects on storm tides. The open boundaries of the nested model are set using Chapman conditions for the free surface, Flather conditions for 2D momentum, and radiation conditions for 3D momentum, and the gradient condition is applied for salinity and temperature, effectively holding them constant within the domain (Marchesiello et al., 2001). Since the spin up is only for velocity and water level, it only requires hours to stabilize."

The information on the initialization of the parent model was available on lines 159-160 of the original manuscript.

*14. L156-157: the authors should locate these gauges on a Figure. The reader should not wait until the results to find the gauge. The same occurs with other gauges mentioned in the methods section.*

Change made. A figure showing the location of the Kings Point, Bergen Basin, and Sandy Hook gauges added. (Figure 2c)

*15. L158: if you are using coastal gauge water levels to represent storm surge conditions, then you are not modeling storm surge directly on your model with a wind and barometric field pressure. This isn't very clear to me and will not be clear to the reader. Please rephrase. Also, if the answer to my question is true, the authors are not modeling storm surge directly; rather, they are just propagating a coastal flood inland. This will need clarification throughout the entire manuscript to change what you call "simulating storm surge" to "simulating coastal flood." The reader will expect that if you are simulating storm surge, then you are applying wind stress to the ocean domain in your model and not just a water level at the boundary condition.*

We do use the spatially varying wind stress and barometric pressure in both model domains as atmospheric forcing (as stated in lines 158 and 176 of the original version of the manuscript).

We understand the reviewer's concern of miscommunication, so we added additional explanation about how we impose storm surge in the model: "Additionally, subtidal water levels calculated from observations at the NOAA tide gauges at Sandy Hook (NJ; NOAA station 8531680) and Kings Point (NY; NOAA station 8516945) were added to the model boundaries in New York Bight and western Long Island Sound, respectively. In addition, we simulate the influences of local wind stress and barometric pressure changes on the regional storm. This combination of boundary conditions and in-domain forcing allows for a more accurate representation of both local and regional storm surge effects."

*16. L155-160: Authors should add a figure of the model domain with a clear description of the model boundary conditions. This will add the explanation in this paragraph.*

Change made. A figure of the parent model with a rectangle pointing to the location of the nested model added to figure2, also an explanation of the model boundary condition added to the section: "The open boundaries of the nested model are set using Chapman conditions for the free surface, Flather conditions for 2D momentum, and radiation conditions for 3D momentum, and the gradient condition is applied for salinity and temperature, effectively holding them constant within the domain."

17. *Figures 2-3: combine them into a single figure with two subpanels. There are many similar instances of this, such as Figures 4-5. Please try to achieve this since the fewer figures, the better for the reader.*
Change made. Figures merged.

18. *L170-174: would this mean that you only simulated pluvial flooding and not compound flooding since your coastal flood was minor during this event? If true, the authors should be more clear on their terminology and manuscript language. The reader will expect a "good amount" of both flood drivers to call the event a compound flood. Please be more clear with this earlier in the manuscript. I want to say that the authors should have selected a different event, such as Hurricane Sandy since Ida did not bring significant coastal floods. Furthermore, the justification of selecting Ida as the event due to data for calibration is not supported since you cannot accurately calibrate a whole watershed with only a handful of high-water marks, especially if they are in places where water does not accumulate greatly. The reader will question why you did not model Sandy, so please include a justification for this and why Ida was best. If your purpose was to simulate extreme rainfall, Ida is fine, but if you wanted to simulate a compound flood event, then Ida was not ideal.*
We have clarified our study's focus and terminology more explicitly, making edits to the beginning of the final paragraph of the introduction, stating:

> This study improves COAWST to enable pluvial and compound (pluvial-coastal) flood simulations. We simulate Ida's flooding in the Jamaica Bay watershed of New York City in 2021, including exploring different counterfactual scenarios of a shifted storm track causing more intense rainfall and a shifted storm timing causing the rain to peak at high tide to cause compound flooding.

While Ida primarily caused pluvial flooding, we included these tests to demonstrate COAWST's ability to simulate compound flooding under hypothetical scenarios of temporal shifts. This capability is a valuable outcome of our study, and we have already stated in the original manuscript that Ida was primarily a pluvial event (line 16), and exploring the potential compound (lines 21, 75).

However, we understand the reviewer's concern, and we have now added further clarification in the abstract (and title) to avoid any potential misunderstanding: "Sensitivity analyses are used to study the broader risk from events like Ida (pluvial) and potential compound (pluvial-coastal) flooding."

Regarding the selection of Ida over Hurricane Sandy, the latter was also not a compound flood event, as less than one inch of rainfall occurred, and it was not raining around the time of peak storm surge. Instead, we have chosen Ida as the primary focus because our goal was to develop the capability and simulate the impact of extreme rainfall. While Ida did not result in significant coastal flooding, the event provided valuable data for calibrating the model's rain and drainage capabilities, which was essential for our research objectives. We added another less intense rainfall event (Sep 29, 2023, with more observed flood values as HWMs) that we simulated and used for validation, described in the supplementary material.

We hope these revisions address the reviewer's concerns and make the study's objectives and methods more transparent.

19. *L173: quantify the adjective "far below". Similarly, the authors have many vague adjectives that need quantification, such as "significant," "accurate," and "good." What for the authors is far below could be different to me.*
Change made. The street level of 3.17 m NAVD88 at the location of Bergen Basin gauge added to the sentence.

20. *L180-186: I consider this too much detail about the radar rainfall source. The authors should use only one sentence and cite a reference unless they extensively use rainfall for their model, which is not the case, I believe.*

Change made. We summarized the section by stating only the justification of using MRMS QPE product:

> "The MRMS QPE data provide 1.11 km spatial and 1-hour temporal resolution, combining radar, satellite, and rain gauge observations with bias correction to offer more accurate precipitation estimates than radar-only products (Zhang et al., 2016)."

21. *L189: how does this 70 mm/hour of rainfall compare with other events at the basin? For example, the authors could compare this with return period values from the NOAA Atlas 14.*

Change made. 70 mm/hour rain intensity or 2.76 inch/hour corresponds to a rain event with 50-year return period based on NOAA Atlas14 at 'NEW YORK JFK INTL' station. The explanation added to the section is:

> "The maximum hourly rainfall intensity was 70 mm/hour during Ida, which is a 50-year return period rain event (2.76 inch/hour) based on NOAA Precipitation Frequency Data Server (station ID: 30-5803) (National Weather Service))."

22. *L192-193: remove this sentence.*

Thank you for your feedback. We have removed the sentence from the main text as requested. To satisfy the journal's policy on proper attribution of data sources, we have retained the information in the figure captions stating that "base maps are from MATLAB, hosted by Esri®". This ensures that the source of the basemaps is clearly acknowledged in the relevant figures.

23. *Figures 4-5: include the basin boundary on the map so we can assess visually the amount of rainfall that falls inside.*

Change made. The watershed boundary added to the figure.

24. *Figure 6: Why show six days of rain if everything happens on a single day? I consider even the graph unnecessary since it gives the impression that the authors are simulating uniform rainfall using the values on this graph.*

Thank you for your feedback. The five-day range is included to show the concentrated nature of the rainfall event and to avoid any implication of continuous or uniform rainfall across the period. We added this to clarify this point in the text to avoid any potential confusion:

> "Figure 4 shows the time series of rainfall to highlight the sharp peak of rainfall during the 5-day simulation and the absence of significant rainfall before and after."

Indeed, we already stated about the using the spatially and temporally varied rain in line 127 of the original manuscript ("The rain rate is included as an additional spatially and temporally varying meteorological forcing variable") to avoid any confusion.

25. *L212: what do the selected values of drain rate physically mean for the system? Do they have any physical justification for being selected, or are they just random values that work?*

The drainage rates represent the infiltration and stormwater system in the model. Candidate values ranged all the way up to 44 mm/hour, the storm water system capacity stated by NYC. Here in the manuscript, we present some values that lead to the best results to match the observed data. We added more clarification in the methods about the process. Also, the discussion on section 4 paragraph five is about the reasons why the system may not work at its design criterion.

26. *L221: all of the urban areas the same? The authors are using a typical value of CN*

*for their whole area. However, areas do change, and the soil type and antecedent conditions also play a role in the curve number value selected. The authors should talk more about this since it is crucial for their results, and not just say everyone uses 90 and that it. I strongly recommend the authors compute their own weighted average CN for their watershed using all the parameters needed since that would be more defendable than the current approach. In my opinion, the authors have been too simple on the hydrologic side of the study, which is crucial for any compound flood simulation.*

We appreciate the reviewer's feedback and have updated this calculation. Instead of applying a uniform CN value of 90, we recalculated a weighted average CN of 93.95. This was determined by using CCAP land cover data, which was already provided in our paper, and soil classification data from the USDA-NRCS Web Soil Survey and the Jamaica Bay Coastal Zone Soil Survey. The soils in the area primarily fall under Hydrologic Soil Group C, reflecting moderate infiltration rates. This updated CN calculation provides a more accurate estimate of the region's runoff potential during Hurricane Ida. We added these data to the manuscript:

> "To reflect the spatial variability in urban and non-urban areas, a weighted CN of 93.95 is derived based on the land cover categories (C-CAP data, Figure 2d) and soil data from USDA-NRCS Web Soil Survey (U.S. Department of Agriculture; Cronshey, 1986)."

27. *L222: Can the authors confirm the rainfall amount in 3-hr, using rain gauges within the basin instead of only depending on the radar data, which will not always be 100% accurate? For example, authors could use the CoCoRHAS network.*

Thank you for your suggestion regarding the use of rain gauges. We compared data from the Howard Beach station (40.66, -73.84) which is in our domain from CoCoRaHS network on 02 Sep 2021, which recorded 126 mm (4.96 inches) of rainfall (7 am), to MRMS data, which reported 102 mm (4 inches) at the same location same day. This demonstrates the reasonable accuracy of MRMS data. It should be noted that the 74 mm (2.93 inches) used in CN calculation is the average rainfall depth per cell across the domain, essential for runoff estimation. Figure 3.b shows spatial variability of the rainfall, with the western domain receiving more rain, explaining why the average is lower than the observed rain at Howard Beach.

We also added this in the text to avoid any misunderstanding:

> "We consider the rainfall associated with Ida as a concentrated 3-hour period of precipitation (as vast majority of the rain is over three hours according to MRMS), starting at 02 Sep 2021, 00:30 UTC. A comparison of the gauge data from the CoCoRaHS at Howard Beach station (126 mm) with MRMS data at the same location (102 mm) confirms the reasonable accuracy of MRMS. During this period the rainfall generates an average of 74 mm (2.9 inches) total accumulation across the domain."

28. *L225-226: this is another big limitation that is not addressed in the manuscript. There are many models out there that use the spatially varying rainfall to generate spatially varying runoff instead of just a single runoff volume for the entire basin. For example, how would the results change if the authors used the average rainfall over that basin instead of the maximum value? Those are key questions that need to be answered within the sensitivity analysis to justify the approach.*

Thank you for your comment. We would like to clarify that the presented runoff calculation using the average rainfall depth across the domain is represented only to compare the model's uniform drain rate (which was calibrated based on the best fit with observed HWMs) with an estimate of potential runoff generated by the storm, as mentioned in line 214 of the original manuscript. It is true that the CN approach is simplistic, but it is only used to provide a simple comparison to the drainage rates. We disagree that this is a limitation in the manuscript, as the manuscript provides much more complex modeling than the CN computation.

29. *L226-227: While I agree with the authors' statement, it is not appropriate for their study due to the gross simplification of uniform rainfall and drainage rate. Designing stormwater infrastructure requires more than this since as you keep going downstream towards the outlet, your main sewer line will start receiving more runoff than upstream or other collecting pipes with a smaller catchment area.*

As noted above (#28), the presentation of the CN is presented just for context and comparing our calibrated drainage rate.

This paper is not giving any suggestions on the stormwater system, we are here only providing a flood map and exploring some possible sensitivity scenarios.

As noted in the original manuscript (Figures 4 and 5), our modeling applies spatially varied rainfall.

30. *L234: if you only shifted the rainfall field, means that you did not shift the wind field. Thus, you compound simulation is not well represented. Furthermore, this statement confirms for me that you did not simulate storm surge over the domain as my previous comment suggests.*

We are shifting rainfall field and wind field when investigating the spatial shift tests. We added "rain fall and wind field" to be more clear about this.

31. *L238-240: this information should be in section 2.1. Also, why not use the return period values at NYC and use the Northeastern states? A local comparison could benefit since a 10-yr return period is not extreme.*

Change made. Looking at the local station also shows the same 3-hour rain with a 10-year return period (71 mm), and if we look at the maximum intensity (70 mm/hour) which already is stated in section 2.3, this corresponds to a one-hour rain with 50-year return period.

We added (section 2.1):

"Ida's rain averaged across the Jamaica Bay watershed was 71 mm (2.8 inches) over 3 hours, which corresponds to a 10-year return period (between 5 and 25 years) based on NOAA Precipitation Frequency Data Server (Station ID 30-5803). However, the maximum intensity (Figure 3) reached to 70 mm/hour, or a one-hour rain event with a 50-year return period (2.76 inch/hour) based on the same NOAA data (National Weather Service)."

32. *Figure 7: combine with Figures 4-5.*

Change made. Figure combined to Fig. 3c.

33. *L257: why not synchronize the peak storm surge with the high tides also? You already had low storm tide conditions. The suggestion made by the authors would not affect your compound flood since the storm tide conditions were already low. This can be shown in the results. If the author manipulates the water level boundary conditions to coincide the peak storm surge with high tide, then they could have a more significant compound flood in the model than what they have now.*

Thank you for your suggestion. We altered the simulations and now we synchronize the peak storm surge with the high tide when investigating the temporal shift cases. As you can see in the figure below, the synchronization of the storm surge with high tide, result in 0.3 meters higher water elevation at Bergen basin gauge.

[Figure]

However, we should mention that now with more HWMs and the additional rain event (29 September 2023), the calibrated drainage rate for the model equals to 13 mm/hour, so this addition in the drainage rate counter acts the higher water elevation. That explains why, despite the evidence of compound flood, its impact is not too severe.

34. *L268: are you referring to the "no infiltration" to the "drainage rate" component of the research? The authors never computed infiltration in the method section. I know that, in theory, it is the same sink term, but this could confuse the readers, so be consistent with your terminology.*
    Change made. Used "drainage rate" instead.

35. *L269: it is technically impossible to see the flood depths to vary spatially since you apply uniform rainfall, drainage rate, and runoff. What you see is a deeper flow in low-lying areas since it tends to accumulate, but a varying rainfall, drainage rate, and runoff (which is more realistic) will show varying floods.*
    Thank you for your insightful comment. We would like to clarify that our model does indeed incorporate spatial and temporal variability in rainfall (Line 127 of the original manuscript) and as a result, runoff. In the base model used to simulate Ida's flood impact on Jamaica Bay, rainfall is not applied uniformly. Instead, we account for temporal and spatial variation in rainfall distribution. This variability, along with the interaction between rainfall and topography, results in the spatially varying flood depths observed in the model output. We have updated the text in the manuscript to clarify this point.

36. *L270: what do the authors mean by Dcrit?*
    For wetting and drying, the minimum depth to allow flow out of the cells, is named Dcrit. This explanation is stated in Line 151 of the original manuscript.

37. *L271: why do the authors label the high-water marks as empirical? Are they were not field surveyed? Explain more.*
    Change made. They are indeed surveyed high-water marks; in the text we used "surveyed" instead to make this clear.

38. *L273-275: remove the sentence since this is obvious when comparing data to high- water marks.*
    Change made. Sentence removed as reviewer's suggestion.

39. *L275: the location of the high-water marks with respect to the deeper flood locations is another limitation of the study that the authors should highlight in the corresponding section.*

The limitation regarding the location of the high-water marks (HWMs) with respect to the areas of maximum inundation is acknowledged in the study. It is important to note that the HWMs used in this analysis represent all the available data. While these HWMs may not perfectly correspond to the locations of maximum modelled inundation, they still provide valuable information for validating the model, albeit with this noted limitation. To address this limitation, further investigation of another rain event (September 29th, 2023) has been added to the calibration and more information about the event itself is presented in the supplementary material. This way by adding another storm and more HWMs, in different areas of the watershed, we compensate for the data limitation and strengthen the overall validation process. We also updated the calibration section, using the HWMs from both events, in the main manuscript.

40. *L286: I would not call what the authors did a hydrologic model.*

"hydrological" removed based on reviewer's suggestion.

41. *L287: Why did the authors select these drain rates? Justify and put into context what these values might represent. For example, an X-in pipe is flowing full. That would make it easier to relate to the real world.*

The selected drain rates of 6 mm/hour (0.25 inch/hour), 13 mm/hour (0.5 inch/hour), and 19 mm/hour (0.75 inch/hour) were chosen based on physical reasoning and calibration against observed data. We agree that adding context would be of value in the paper, so added the line to Section 2.4: "Potential drain rate values extend up to 44 mm/hour, which corresponds to the design capacity of the stormwater system (Nyc-Mocej, 2023)." The 13 mm/hour rate, which provided the best fit to high-water marks from both the main event and a less intense event (details in supplementary material), reflects a reasonable drainage rate for the model based on the best fit with observed HWMs.

42. *L296: the authors mention here that the Cn calculations were used using hourly rainfall, but in the respective section, they said they use the maximum value of rainfall over the basin within a 3-hour time window. Therefore, these arguments confuse the reader. Please decide what hourly runoff or a single value you used.*

Change made to add clarity: "According to the CN calculations in Section 2.4, Hurricane Ida is estimated to generate an average of 58 mm of runoff over a 3-hour period, which corresponds to a runoff rate of approximately 19 mm/hour (or 0.75 inch/hour). This establishes the necessary rate at which stormwater must be managed to ensure proper drainage."

43. *L305: what do the authors mean by "non-zero depth HWMs"?*

This refer to the HWMs that are not zero-depth in the model output (in the plots) but remain unchanged with drainage rate variation. We changed the sentence to be more clear: "The HWMs (in Figure 7 plots) that remain unchanged with increasing drainage rates are elevated, steeply sloping locations where water rapidly flows and does not accumulate."

44. *L314-317: can you reference someone else who has followed a similar flood classification? Also, why is important to classify them as this? Also, I would expect the authors to change the color scale on the results to use this classification rather than the numeric values. If not, then what is the purpose of the classification in the first place?*

Thank you for your insightful feedback on the flood depth classification and its relevance to our study. The purpose of the simplified classification is to describe impacts, much like National Weather Service metrics of flooding, minor-moderate-major.

While the classification system used in the paper (shallow: 0–0.3 m, deep: 0.3–0.9 m, and extreme: > 0.9 m) is not an official standard, it is grounded in practical observations of flood impacts. This kind of classification simplifies communication, helping readers quickly understand the varying levels of risk associated with different flood depths.

Regarding your comment about using the classification in the color scale of the results, we chose to present the data in its continuous form in the figure. This allows us to convey the detailed variation in flood depths rather than reducing it to broad categories. While the classification is useful for summarizing and communicating results in the discussion, the figure serves a different purpose figure has the different purpose to convey the data in more detail.

*45. Figure 11: why not include the histogram insert in Figure 9? Also, the authors should combine this figure with Figure 9.*

Thank you for your suggestion. Histograms are presented only for the calibrated model (simulation with a 0.5-inch/hour drainage rate) and are shown for both the flood depth map (as seen in Fig. 11 of the original manuscript) and the speed map. Figure 9 (in the absence of drainage rate) in the original manuscript shows the HWM locations and helps readers visually compare the effect of drainage rate on the spatial variation of flood depth. Given this purpose, we believe it is useful to retain this figure separately rather than combining it with other figures.

*46. L361: it is not clear what is the second scenario within the context of the sentence.*

Thank you for your feedback. Temporal shift scenarios are explained in detail in section 2.5 of the original manuscript (line 259). To add clarity, we have added a reference to section 2.5 in this line to clarify what the second scenario refers to.

*47. L363-364: the authors do not have the necessary results to support this claim. For example, if the authors have run three different flooding scenarios (rainfall only, storm tide only, and compound flood), then they could comment on how important are each of those flood drivers in their modeling approach and in the compound flood assessment.*

Thank you for your comment. Our statement does not intend to imply that we performed a detailed compound flood assessment or quantified each individual flood driver. Instead, we simply stated, "These findings underscore the model's utility in representing compound flooding events," which refers to the counterfactual scenario we presented. We believe this scenario provides valuable insights for both readers and emergency managers. For further details on the model's utility, please see #48 below.

*48. L376-377: I disagree with the authors. While simple models could be useful, the authors fail to show this on the manuscript. Thus, it results does not support the claim.*

We have strengthened the support for this claim by publishing Version 1 of the flood map for Ida, in collaboration with NYC Emergency Management (NYCEM) (Kasaei et al., 2024). They found the map useful for understanding the flooded areas and for internal purposes.

*49. L381: the authors use the concepts of vulnerability and risk throughout the entire manuscript. However, they really only focus on the hazard component. To talk about risk the authors need to talk about the socio-demographics and the community's exposure. I will move away from these concepts since it is not the main focus of the study and replace it with the term hazard.*

Thank you for your insightful comment. As our study focuses primarily on the spatial analysis of flooding hazards in the Jamaica Bay watershed and does not include a detailed exploration of socio-demographic factors or community exposure, we will adjust our language to reflect this.

50. *L388: can this only be because tides? Authors should show results with and without rain and tides to see how each driver interacts, and they could make a similar statement.*

   The reviewer caught an error in the text, and we have revised it to focus on the coastal driver influencing the extreme pluvial event, which is what the paper has quantified: "This indicates that compound flooding can cause greater danger in coastal flood zones during an extreme pluvial event." We also note that we revised the temporal shift simulation to include storm surge, in addition to the high tide and rainfall.  We also revised the figure, so that it shows both the flood depth difference map (left) and the grid cells with additional flood depth (more than 5 cm) around Hamilton Beach (right).

51. *L389: the authors talked about coastal flood zones but failed to define them. I will suggest citing Bilskie and Hagen (2018).*
   Change made to improve the readability of the paragraph.

52. *L391-393: the authors do not support this statement with their current results.*
   We agree that the manuscript doesn't prove the model adequately captures compound flood processes because the model validation is only for a pluvial flood.  Therefore, we have removed the word "processes" and left the remainder of the statement intact. "These results illustrate the capability of the COAWST to capture compound flood effects, and its utility for future modeling of a wider range of coastal-pluvial forcings to improve our understanding of coastal-urban flood hazard."

53. *L396-398: There are already models that exist and are similar to the proposed approach but without the simplification taken here. For example, ICPR (known as StormWise) can include stormwater infrastructure, spatially varying rain, and runoff, and include coastal boundaries similar to the author's approach.*
   In response to the reviewer's comment, we acknowledge that models like ICPR (StormWise) can handle stormwater infrastructure, spatially varying rainfall, and coastal boundaries. However, our approach differs by utilizing COAWST-ROMS, a fully 3D model, which we have improved to incorporate rainfall and drainage rates to represent stormwater systems. This 3D capability allows for more detailed simulation of vertical processes and interactions between stormwater and coastal dynamics, which ICPR, being a primarily 2D model, cannot capture. Our enhancements provide a more integrated framework for modeling complex coastal-stormwater interactions.

54. *L404 and 406: the word "run off" should be together like "runoff".*
   Change made.

55. *L423: the authors should not claim the sediment transport and erosion component of the model since it was not tested.*
   Thank you for your valuable comment. We acknowledge that the sediment transport and erosion component of the model has not been directly tested in this study. We have revised the statement to clarify that this aspect is a potential application of the model rather than a feature that has been validated in this work. The updated text now reflects that our research primarily focuses on urban pluvial flood studies, and the influence of rain on estuary hydrodynamics: "Research made possible by this new model includes infrastructure adaptation planning for urban coastal pluvial flood studies, analyses of rain influence on estuary hydrodynamics, and has the potential for future studies on coupled pluvial-coastal flood-induced sediment transport and erosion."

Dresback, K. M., Szpilka, C. M., Kolar, R. L., Moghimi, S., and Myers, E. P.: Development and validation of accumulation term (Distributed and/or Point Source) in a Finite Element Hydrodynamic model, Journal of Marine Science and Engineering, 11, 248, 2023.

Kasaei, S., Orton, P., Ralston, D., and Warner, J.: Post-tropical cyclone Ida (2021) flood map for New York City's Jamaica Bay watershed (1), Mendeley [dataset], 10.17632/hs2zt6ngwd.1, 2024.

Precipitation Frequency Data Server (PFDS): https://hdsc.nws.noaa.gov/pfds/pfds_gis.html, last

NYC-MOCEJ: New York City Stormwater Resiliency Plan, Report, 2023.

Santiago-Collazo, F. L., Bilskie, M. V., Bacopoulos, P., and Hagen, S. C.: Compound inundation modeling of a 1-D idealized coastal watershed using a reduced-physics approach, Water Resources Research, 60, e2023WR035718, 2024.

Zhang, J., Howard, K., Langston, C., Kaney, B., Qi, Y., Tang, L., Grams, H., Wang, Y., Cocks, S., and Martinaitis, S.: Multi-Radar Multi-Sensor (MRMS) quantitative precipitation estimation: Initial operating capabilities, Bulletin of the American Meteorological Society, 97, 621-638, 2016.

---

## Author Comment (AC2)

The authors thank the reviewers for a very detailed reading of the paper and substantive comments that have clearly improved the research and its presentation. Below are the original comments in blue color and italics and our responses below them.

**Anonymous Referee #2**
*Received: 04 October 2024*

**Summary**

The manuscript provides a modeling study to simulate the effects of Ida on pluvial flooding in New York City's Jamaica Bay watershed. The major advancement is that the authors parameterize soil infiltration and a stormwater conveyance system as a drainage rate, which shows improved model performance when compared against high water marks. The authors also performed a sensitivity study by shifting the storm tracks and the timing of rainfall. Overall, the article is well written. The results are reasonable. I have a few concerns and hope the authors could address them.

**General comments**

1. *My primary concern is the simplified approach used for modeling urban drainage, which, while practical, presents several limitations. A notable drawback is that the single-parameter drainage rate requires calibration specific to an event and lacks generalizability, making the method less practical especially when there is limited data for calibration. The method also neglects detailed factors like varying land cover and the complexities of urban drainage systems. The authors should provide a more compelling rationale for adopting this method over more detailed urban stormwater models.*
   We acknowledge the concern regarding the simplified approach used for urban drainage modeling. The decision to adopt this method was influenced by the challenges involved in assembling a detailed hydrologic and hydraulic (H&H) model, as highlighted in the paper. Currently, no comprehensive simulation of Ida for the area in question exists, and we are collaborating with NYC-Emergency management and published Version 1 of the Ida flood map(Kasaei et al., 2024).

   However, progress is being made to address the issue of calibration data scarcity. The simulation conducted on September 29th, 2023, illustrates how more data is becoming available, particularly with the ongoing deployment of 500 sensors as part of the Floodnet.nyc project. This project is expected to significantly improve data availability, paving the way for more refined model calibrations (e.g. with rain-rate dependent drain rates), and enhancing the generalizability of future simulations.

2. *The sensitivity experiments that shift storm tracks and timing appear over idealized. Altering storm tracks should realistically affect the intensity and spatial distribution of rainfall, among other storm characteristics. Although it may be challenging for the authors to accurately capture these changes, it remains crucial for them to justify their approach and discuss the associated limitations and uncertainties.*
   We agree the sensitivity experiments are simplistic, but they serve the purpose of demonstrating our model and exploring potential worst cases in spatial and temporal storm variations. For both spatial and temporal scenarios, the wind also shifted along with the rain. Also, we have now improved the temporal shift by also shifting the storm surge.
   We have modified the text to acknowledge this simplicity, stating:
   > "These experiments are simplistic and true variations and uncertainties in storms can affect a wide range of storm characteristics including intensity and spatial distribution of rainfall. However, a comprehensive study of Ida forecasting uncertainties is beyond the scope of this paper."

3. *The manuscript focuses solely on the impact of rainfall, which is suitable for studying pluvial flooding. However, the title and various sections refer to compound flooding. It remains unclear how compound flooding, particularly in relation to the co-occurrence of storm surge, high tide, or coastal*

*flooding, is relevant to this study. The manuscript briefly mentions these factors but does not adequately explain how the study's findings apply to scenarios involving compound flooding.*

We appreciate the reviewer's feedback regarding the title and scope of the study. We acknowledge the importance of ensuring the manuscript accurately reflects its focus. Our study primarily aims to simulate extreme pluvial flooding caused by Hurricane Ida, which, as highlighted in lines 16, 21, and 75 of the manuscript, was primarily driven by heavy rainfall. As such, our core objective is to enhance the COAWST model by incorporating rain and drain rates to better simulate such rainfall driven events. While we did explore the potential for compound flooding through sensitivity analyses, the primary focus of this study remains on pluvial flooding. To better clarify this and address the concern, we have revised the manuscript's title to more clearly reflect the emphasis on pluvial flooding. We have also made clear in the Introduction that potential compound flooding scenarios were considered through sensitivity tests. This will ensure the title and introduction more accurately align with the study's scope.

**Specific comments:**

1. *P5L130. "The drain rate can be a negative when it is locally greater than the rain rate." This is reasonable. But is there a limit for the range of drain rate?*

   Change made to make the sentence clearer:

   > "The drain rate is always negative (a volume sink representing stormwater system and infiltration) while the rain rate is always positive (a volume source), and the net rate of volume change (precipitation-drain rate) can be negative when it is locally greater than the rain rate."

2. *Section 2.2.2 A bit more details on the model setup would be very helpful. For example, how are the two models nested? A zoomed map showing the high-resolution grid on top of the larger scale grid would help reader understand the bigger picture.*

   Change made. We added explanation of the nested model boundary condition in section 2.2.2:

   > ("The open boundaries of the nested model are set using Chapman conditions for the free surface, Flather conditions for 2D momentum, and radiation conditions for 3D momentum, and the gradient condition is applied for salinity and temperature, effectively holding them constant within the domain.").

   in addition, we added a figure showing the regional model and the nested model boundaries inside it (Fig. 2a).

3. *P7L175. Figure 8 is referenced earlier here. The authors may correct the order of figures.*

   Thank you for your insightful comment. We acknowledge that Figure 8 is referenced earlier in the text, specifically regarding the coastal water level range during the simulation period in Jamaica Bay. We chose to reference Figure 8 sooner due to its relevance to the water level range, even though the figure appears later in the paper as it is about temporal shift of the storm. We believe this structure better supports the flow of the discussion, but we will ensure that the figure numbering and placement are consistent throughout the manuscript.

4. *P10L230 "buy"*

   Change made. Thank you so much for noting it.

5. *P12L268. "base model". Since you are running one model, "baseline simulation" may be more accurate.*

   Change made. Used "baseline simulation" instead.

6. *P12L268. "infiltration, no spatial or temporal shifting of rain, no temporal shifting of rain", repeated expression.*
Change made. Removed the repeated expression.

7. *P12L277. "This discrepancy". Could it be other reasons? Such as the uncertainty in the atmospheric forcing?*
We appreciate the reviewer's suggestion. While our analysis primarily attributes the discrepancy to the omission of infiltration and stormwater drainage, we agree that uncertainties in atmospheric forcing (e.g., wind, precipitation) could also play a role. We will acknowledge this in the text:

"This discrepancy is likely attributed to the model's omission of infiltration and storm water drainage; however, it is important to acknowledge that other factors, such as uncertainties in atmospheric forcing, could also contribute to this discrepancy."

8. *P12L281-283. I would recommend providing full time series of such validation results, either provided here or in the supporting information.*
Change made. We added the time series as an additional figure as the reviewer suggested in the supplementary material.

9. *P14L297-298. This is a bit confusing. Please consider rephrasing.*
We have now rephrased the text to avoid confusion:

"According to the CN calculations in Section 2.4, Hurricane Ida is estimated to generate an average of 58 mm of runoff over a 3-hour period, which corresponds to a runoff rate of approximately 19 mm/hour (or 0.75 inch/hour). This establishes the necessary rate at which stormwater must be managed to ensure proper drainage."

10. *P16L333-334. This is true. It seems that many areas have flow speed over 1 m/s. And the top speed of 4 m/s seems too high. Is this reasonable for urban flooding? Also, I think the grid resolution cannot resolve streets. When you zoom in, it may be more clear to look at the spatial patterns of flow speed.*

We appreciate the reviewer's comment. Regions with flow speeds over 1 m/s generally occur in areas with steep slopes around 20 degrees (see Figure 2 for the DEM). For flow speeds greater than 3 m/s, the slopes tend to be even steeper. However, as noted in the limitations section, the use of bare-earth DEM may underrepresent buildings and streets, which could impact flow speed accuracy in some areas. This improvement could be addressed in the future to make it feasible to analyze the spatial patterns in more detail.

11. *Section 3.3.2. The results are only superficially mentioned here. More figures and discussions should be provided.*
Thank you so much for your suggestion. We added one more figure and discussion as reviewer suggested.

12. *P19L407-409. This is a bit confusing. Please elaborate.*
While our model and the curve number method use hourly-accumulated rainfall data, short duration rain bursts likely exceeded the system's capacity. For example, intense rain within a short period (less than an hour) would produce much higher runoff rates than the hourly average suggests. These peaks likely overwhelmed the system, contributing to flooding. This also explains why our model's calibrated drain rate of 6 mm/hour is lower than the design stormwater capacity of 44 mm/hour, as short bursts can create momentary runoff much higher than the hourly accumulation captures.

We revised the sentence to make it more clear:

> "However, we should consider that it is likely that brief, intense rain bursts, rather than the hourly MRMS rain rates that were used in this study, caused most of the flooding by overwhelming the stormwater system."

**Figures:**

*13. Figure 1 is a bit confusing as it is difficult to distinguish between land and ocean. Blue contour is also confusing. The authors may also consider showing the watershed boundary.*
Change made. The watershed boundary added to the figure.

*14. Figure 11 Magnified views at selected regions will be helpful to interpret the modeled flooding.*
Thank you so much for your feedback. As we mentioned in the original manuscript, we are using a bare earth DEM and according to the grid resolution we do not resolve most street valleys, only the largest ones. As such, although the model shows the spatial variability of flood map for Ida, zooming into the streets may cause confusion as we do apply the model on a bare earth DEM.

*15. Figure 14, Is the difference only presented in this region? A figure with a greater extent and zoomed views may help interpret the results.*
Change made. We added the difference flood map for second scenario of temporal shift, and the zoomed in panel to help the reader to understand better. We do see some other area with additional flood depth in east and west side of the bay. We are zooming in on the Hamilton Beach area as an example.

Kasaei, S., Orton, P., Ralston, D., and Warner, J.: Post-tropical cyclone Ida (2021) flood map for New York City's Jamaica Bay watershed (1), Mendeley [dataset], 10.17632/hs2zt6ngwd.1, 2024.

---

## Author Comment (AC3)

The authors thank the reviewers for a very detailed reading of the paper and substantive comments that have clearly improved the research and its presentation. Below are the original comments in blue color and italics and our responses below them.

**# Anonymous Referee #3**
*Received: 08 October 2024*

1. *The title mentions both pluvial and compound flooding, yet the study primarily focuses on pluvial flooding. The scope of compound flooding, including the interaction of coastal drivers like storm surge, is not adequately addressed. The title and introduction should better reflect the actual scope of the study.*

   We appreciate the reviewer's feedback regarding the title and the scope of the study. We acknowledge the need to better align the title and introduction with the study's focus. The primary objective of our study is to simulate the extreme pluvial flooding caused by Hurricane Ida, which, as noted in the original manuscript (lines 16, 21, 75), was primarily a pluvial event, and also to improve COAWST to incorporate rain and drain rates. While we explored potential compound flooding through sensitivity analyses, the primary focus remains on pluvial flooding. To reflect this more clearly, we have revised the title to make the study's scope clearer while acknowledging the potential for compound flooding considered in our sensitivity tests.

2. *The single-parameter drainage rate approach, while practical, limits the model's generalizability and accuracy across different events. The authors should provide a stronger justification for this method, considering more detailed models that include spatially varying land cover and urban infrastructure.*

   We acknowledge the limitation of using a single-parameter drainage rate. While this approach simplifies the model, it has shown to be effective in replicating flood patterns during intense rainfall events, as demonstrated by the alignment of our model outputs with high-water mark (HWM) observations. Additionally, the model can incorporate spatially varying drainage rates based on storm water system capacity and can be refined in future versions to improve generalizability across different events, as mentioned in the future work section.

3. *The sensitivity experiments involving shifting storm tracks do not account for changes in rainfall intensity or spatial distribution. This idealized approach oversimplifies the relationship between storm track variation and its impact on pluvial flooding. A discussion of these limitations should be included.*

   We agree the sensitivity experiments are simplistic, but they serve the purpose of demonstrating our model and exploring potential worse cases in spatial and temporal storm variations. Also, we have now improved the temporal shift by also shifting the storm surge. We have modified the text to acknowledge this simplicity, stating:

   > "These experiments are simplistic and true variations and uncertainties in storms can affect a wide range of storm characteristics including intensity and spatial distribution of rainfall. However, a comprehensive study of Ida forecasting uncertainties is beyond the scope of this paper."

4. *The manuscript's discussion of compound flooding is unclear, particularly in the context of storm surge, high tide, and coastal interactions. These aspects are not fully explored in the results, and the study seems to focus solely on pluvial flooding. The authors should either adjust the scope or provide more detailed analysis of compound flood scenarios.*

   We appreciate the reviewer's feedback regarding the detailed discussion on compound scenarios. We improved the compound flood scenarios context by adding a figure, an explanation of how the scenarios were conducted, and more discussion of the results.

5.  *The need for event-specific calibration for the drainage rate method poses a challenge for practical application, particularly in data-scarce regions. More discussion is needed on the calibration process and how it could affect the model's reliability in broader applications.*
    We value your concern about the calibration process, and the scarcity of the available observed data for validation as we acknowledged in the original manuscript (line 435). To improve the robustness of the validation procedure, we add another rainfall event (29[th] September 2023), which we have more HWMs available for calibrating the model. More information about the event is added as a supplementary and the comparison plots have updated to include all data points available on both storms. This way we believe the calibration is more general, and with future increase in data availability (as mentioned in line 435 of the original manuscript), this robustness can be improved dramatically. In the original manuscript Line 411, we noted that according to the calibration and drain rates, our model may be more applicable to the extreme rain events and likely has lower accuracy or need for new running for weaker rain events.

6.  *The claim that urban flood flow speeds reach up to 4 m/s seems high for the grid resolution used. The authors should justify this finding with a discussion on the limitations of their spatial resolution in capturing detailed street-level flow dynamics.*

    We appreciate the reviewer's comment. The observed flow speeds over 1m/s are found in areas with steeper slopes around 20 degrees (see Figure 2 for the DEM). For the flow speeds more than 3m/s, the regions have even steeper slopes. However, we acknowledge that the grid resolution used may not fully capture street-level details, as noted in the limitations section ("the moderate ~50-meter spatial resolution may overlook finer-scale variations in flooding. Potential future improvements to mitigate these limitations are discussed in Sect. 4.").

7.  *The authors should provide more context on how these values were selected and their physical relevance, especially in representing urban stormwater systems.*
    The selected drain rates (6 mm/hour, 13 mm/hour, and 19 mm/hour) are based on physical reasoning and model calibration using high-water marks (HWM) from observed storm events (details in the supplementary material). Basically, the candidate values ranged all the way up to 44mm/hour (storm water system capacity of NYC), and the ones that lead to the best results are presented in the manuscript. We added this explanation to the manuscript as well.

8.  *The authors should discuss the implications of using such limited number of high-water marks for model validation, particularly in relation to spatial variability in urban flooding.*
    We acknowledge the limitation of using a small number of high-water marks (HWMs) for model validation, especially regarding spatial variability in urban flooding. While the HWMs used represent all available data from Hurricane Ida, their spatial distribution may not fully capture the areas of maximum inundation. To mitigate this limitation, we included an additional less intense rainfall event from September 29, 2023, in the supplementary material, which provided more observed flood values (HWMs) and allowed for a more comprehensive validation. This approach helps strengthen the overall assessment of the model's performance across different spatial and rainfall conditions.

9.  *The authors mention a 70 mm/hour rainfall rate but do not provide context on how this compares to return period values (e.g., from NOAA Atlas 14). A comparison to historical rainfall events would provide readers with a clearer understanding of the extremity of the event.*
    Change made. 70 mm/hour rain intensity or 2.76 inch/hour corresponds to a rain event with 50-year return period based on NOAA Atlas14 at 'NEW YORK JFK INTL' station. The explanation added to the section: "Ida's rain averaged across the Jamaica Bay watershed was 71 mm (2.8 inches) over 3 hours, which corresponds to a 10-year return period (between 5 and 25 years) based on NOAA Precipitation Frequency Data Server (Station ID 30-5803). However, the maximum intensity (Figure

3) reached 70 mm/hour, or a one-hour rain event with a 50-year return period (2.76 inch/hour) based on the same NOAA data (National Weather Service)."

10. *The boundary conditions for storm surge are not clearly explained. If the authors are only simulating coastal flood propagation without directly modeling storm surge, this should be clarified to avoid misleading the reader.*
We do use the spatially varying wind stress and barometric pressure in both model domains as atmospheric forcing (as stated in lines 158 and 176 of the original version of the manuscript). We understand the reviewer's concern of miscommunication, so we added additional explanation about how we impose storm surge in the model:
   "Additionally, subtidal water levels calculated from observations at the NOAA tide gauges at Sandy Hook (NJ; NOAA station 8531680) and Kings Point (NY; NOAA station 8516945) (Figure 2c) were added to the boundaries in New York Bight and western Long Island Sound, respectively. In addition, we simulate the influences of local wind stress and barometric pressure changes on the regional storm. This combination of boundary conditions and in-domain forcing allows for a more accurate representation of both local and regional storm surge effects."

11. *The use of uniform rainfall across the watershed is a significant limitation. A more detailed analysis using spatially varying rainfall inputs, or a discussion on how this uniform approach affects model results, would improve the manuscript.*
We would like to clarify that our model does indeed incorporate spatially and temporally varied rainfall inputs, as mentioned on lines 127 and 179 of the original manuscript.
These inputs were specifically chosen to reflect the variability in rainfall across the watershed to enhance the accuracy of our model results.

12. *The manuscript claims to model storm surge but does not appear to include wind stress or pressure fields. This limits the realism of the coastal component of the flood model. A discussion on how these factors could be incorporated would strengthen the study.*
We would like to clarify that our model does indeed incorporate spatially varied wind stress and pressure fields as part of the meteorological forcing, which we have already mentioned in line 176 of the original manuscript.

13. *The manuscript uses a single CN value (90) for the entire urban area, which oversimplifies land cover and antecedent conditions. The authors should compute a weighted average CN or provide more justification for using a uniform value.*
We appreciate the reviewer's feedback and have updated our hydrological analysis. Instead of applying a uniform CN value of 90, we recalculated a weighted average CN of 93.95. This was determined by using CCAP land cover data, which was already provided in our paper, and soil classification data from the USDA-NRCS Web Soil Survey and the Jamaica Bay Coastal Zone Soil Survey. The soils in the area primarily fall under Hydrologic Soil Group C, reflecting moderate infiltration rates. This updated CN calculation provides a more accurate estimate of the region's runoff potential during Hurricane Ida. We added these data to the manuscript:
   "To reflect the spatial variability in urban and non-urban areas, a weighted CN of 93.95 is derived based on the land cover categories (C-CAP data, Figure 2d) and soil data from USDA-NRCS Web Soil Survey (U.S. Department of Agriculture; Cronshey, 1986)."

Cronshey, R.: Urban hydrology for small watersheds, 55, US Department of Agriculture, Soil Conservation Service, Engineering Division1986.

Web Soil Survey: https://websoilsurvey.nrcs.usda.gov, last access: 2024.

---

## Referee Report (RR1)

Referee #1

- General Comment #2: Thanks for making your own section of limitations. However, the author has duplicated this statement in the manuscript since the last paragraph of the discussion, which is the limitation in section 2.7, is still in the revised version. Please keep only one. Also, I still think that the authors could include more limitations that are not mentioned in this section (see my original comment in the first revision).

- Specific Comment #16: The authors did not fully understand my request. I do appreciate your detailed response to the boundary conditions. However, the reader would benefit from having a visual representation of the boundaries within the domain. For example, where within the ocean/shoreline in the domain the model is forced with the coastal conditions?

---

## Author Response (AR2)

The authors thank the reviewers for reading the paper again. Below is our response to the minor comments (original comments in blue color and italics and our responses below them).

**# Anonymous Referee #1**
*Received: 01 Feb 2025*

*1. General Comment #2: Thanks for making your own section of limitations. However, the author has duplicated this statement in the manuscript since the last paragraph of the discussion, which is the limitation in section 2.7, is still in the revised version. Please keep only one. Also, I still think that the authors could include more limitations that are not mentioned in this section (see my original comment in the first revision).*

Change made. We removed the duplicate part in section 4.

*2. Specific Comment #16: The authors did not fully understand my request. I do appreciate your detailed response to the boundary conditions. However, the reader would benefit from having a visual representation of the boundaries within the domain. For example, where within the ocean/shoreline in the domain the model is forced with the coastal conditions?*

Thanks for the reviewer's comment. We already provided the visual representation of the boundaries in Figure. 2a. We have made an addition to clarify this, pointing again to the Figure after explaining the boundaries (Line 169 of the manuscript).

The authors thank the reviewers for reading the paper again. Below is our response to the minor comments (original comments in blue color and italics and our responses below them).

**Anonymous Referee #2**
*Received: 02 Feb 2025*

*1. Regarding my earlier comment on the idealized storm tracks and timing, I appreciate the authors' response and the addition of a brief discussion on the associated uncertainty. However, I recommend expanding the discussion on atmospheric uncertainties. For example, Xu et al. (2025) compared different atmospheric forcing products and demonstrated that such uncertainties could be substantial relative to model structure. Similarly, Feng et al. (2024) conducted multiple atmospheric simulations for the same hurricane to account for this variability. See the references below. These uncertainties could significantly impact simulation accuracy, even with the authors' improved modeling approaches (also see my previous comment 7). If the authors choose not to conduct a more in-depth uncertainty evaluation, I suggest elaborating further on this topic in the discussion.*

We followed some of your suggestions and added additional discussion on this topic in the Section 4 (discussion).

*2. Additionally, I previously requested a wider spatial extent and more detailed zoomed-in views for Figure 14 (now Figure 11). While the authors have made revisions, there is still room for improvement. Specifically, in the overall view (Figure 11a), the zoomed-in region is clearly outlined, but the other details remain difficult to discern due to the small size of the dots. Enhancing their visibility would improve the figure's clarity and usefulness.*

We improved the visibility of the Figure. 11a by just showing the differences greater than 5 cm. However, we retained the original dot sizes, as increasing them created a misleading impression of excessive flooding that was not accurate.